# ObjEmbed: Towards Universal Multimodal Object Embeddings

**Shenghao Fu** [* 1 2]  **Yukun Su** [* 2]  **Fengyun Rao** [2]  **Jing LYU** [2]  **Xiaohua Xie** [† 1 3 4 5]  **Wei-Shi Zheng** [† 1 3 4 5]

## Abstract

Aligning objects with corresponding textual descriptions is a fundamental challenge and a realistic requirement in vision-language understanding. While recent multimodal embedding models excel at global image-text alignment, they often struggle with fine-grained alignment between image regions and specific phrases. In this work, we present ObjEmbed, a novel MLLM embedding model that decomposes the input image into multiple regional embeddings, each corresponding to an individual object, along with global embeddings. It supports a wide range of visual understanding tasks like visual grounding, local image retrieval, and global image retrieval. ObjEmbed enjoys three key properties: (1) **Object-Oriented Representation**: It captures both semantic and spatial aspects of objects by generating two complementary embeddings for each region: an object embedding for semantic matching and an IoU embedding that predicts localization quality. The final object matching score combines semantic similarity with the predicted IoU, enabling more accurate retrieval. (2) **Versatility**: It seamlessly handles both region-level and image-level tasks. (3) **Efficient Encoding**: All objects in an image, along with the full image, are encoded in a single forward pass for high efficiency. Superior performance on 18 diverse benchmarks demonstrates its strong semantic discrimination. Code is available at https://github.com/WeChatCV/ObjEmbed.

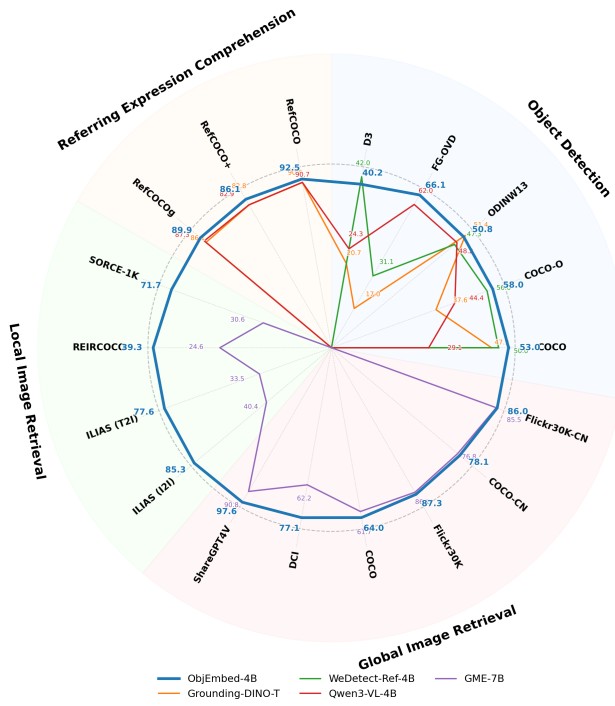

*Figure 1.* ObjEmbed achieves balanced and superior performance across a wide span of benchmarks.

## 1. Introduction

Multimodal embedding models have emerged as a cornerstone in bridging heterogeneous data modalities, such as vision, language, and audio, into a unified semantic space, enabling rich cross-modal understanding, retrieval, and reasoning. Recent advances in large-scale image-text contrastive learning (Radford et al., 2021; Zhai et al., 2023; Xie et al., 2025b) have led to significant progress in multimodal representation learning, especially in aligning images and corresponding captions. Powered by large multimodal models (Bai et al., 2025b;a), embedding models (Zhang et al., 2024; 2025; Jiang et al., 2025c; Lan et al., 2025) can generate task-specific embeddings following user instructions, generate cross-modal embeddings with arbitrary modality combinations, and even perform high-level summary and deep reasoning through chain-of-thought.

However, encoding objects within images and aligning them with text queries are still challenging for recent embedding

*Equal contribution [1] School of Computer Science and Engineering, Sun Yat-sen University, China [2] WeChat Vision, Tencent Inc. [3] Key Laboratory of Machine Intelligence and Advanced Computing, Ministry of Education, China [4] Guangdong Province Key Laboratory of Information Security Technology, China [5] Pazhou Laboratory (Huangpu), China. Correspondence to: Xiaohua Xie <xiexiaoh6@mail.sysu.edu.cn>, Wei-Shi Zheng <wszheng@ieee.org>.

*Proceedings of the 43rd International Conference on Machine Learning*, Seoul, South Korea. PMLR 306, 2026. Copyright 2026 by the author(s).

models. Such capabilities are often critical in real-world applications such as autonomous driving (e.g., distant traffic signs), robotics (e.g., small parts manipulation), and digital content safety moderation. Reliable retrieval and representation of objects demand precise localization and strong semantic discrimination, yet remain under-addressed in current frameworks. While FG-CLIP (Xie et al., 2025b;a) improves regional alignment by combining regional and global contrastive objectives, its object embeddings lack explicit modeling of bounding box quality, limiting their reliability in localization-sensitive tasks. Another line of research, open-vocabulary object detection (Liu et al., 2024c; Fu et al., 2025c;b; Liu et al., 2024a), directly aligning text embeddings with regions of interest, can accurately localize objects and perform open-vocabulary recognition. However, due to limited training data, their generalization ability is largely constrained.

To encode objects with high semantic discrimination and precise localization awareness, we introduce ObjEmbed, an MLLM-based object embedding model that encodes all objects within an image as embeddings. Given an input image, regions of interest (RoIs) are first extracted using an off-the-shelf proposal generator and then encoded as a sequence of tokens. Each object is represented by two special tokens: (1) an object token to capture fine-grained semantic content; and (2) an IoU token to predict the quality of the corresponding bounding box by regressing its IoU score with the ground truth. The object and IoU tokens, along with global image tokens, are processed in parallel by a large language model (LLM) to ensure efficiency. The final-layer hidden states of these tokens serve as the object embeddings, IoU embeddings, and image embeddings, respectively. Similarly, text queries are independently encoded into text embeddings through the same LLM backbone, enabling seamless cross-modal alignment. With this novel architecture, ObjEmbed produces object-centric representations that jointly encode semantic meaning and localization confidence, enabling accurate recognition, precise localization, and robust cross-modal retrieval.

Equipped with this object-centric design, ObjEmbed supports a wide range of downstream applications in a unified framework: (1) Object detection and referring expression comprehension: Class names or natural language expressions are encoded as text embeddings and matched against all object embeddings in the image. The final object matching score combines both semantic similarity (between object and text embeddings) and predicted localization quality (from the IoU embedding), computed as their product, effectively balancing semantic relevance and spatial accuracy. (2) Local image retrieval: When the query describes only a specific region or object, we compute the image-level relevance as the maximum matching score across all detected objects. This strategy enables fine-grained, part-aware retrieval even when the target occupies a small portion of the image. (3) Global image retrieval: Since ObjEmbed also retains global image embeddings, they can be directly used for standard image-text retrieval tasks, ensuring compatibility with conventional benchmarks and applications.

After training on 1.3M samples, ObjEmbed demonstrates strong performance across a wide range of vision tasks, summarized in Figure 1. On object detection, it achieves 53.0% mAP on the COCO dataset, a highly competitive result compared to specialist models. For referring expression comprehension, ObjEmbed attains an average accuracy of 89.5 on RefCOCO/+/g, reflecting its robust ability to align language with visual objects. Most notably, ObjEmbed excels in local image retrieval, which requires fine-grained cross-image comparisons. In this setting, ObjEmbed outperforms existing global image embedding models by around 20 points on four standard benchmarks. It also achieves competitive performance on global image retrieval tasks, despite using a small-scale training set. We hope that ObjEmbed can serve as a strong and general-purpose baseline for future research in object representation learning.

## 2. Related Work

### 2.1. Multimodal Embedding Models

Contrastive language-image pre-training (Radford et al., 2021), which aims to align matched image-text embeddings while pushing away others, is an effective and scalable way to learn transferable image representations. Other improvements, including sigmoid loss (Zhai et al., 2023), well-curated data (Xu et al., 2024; Chuang et al., 2025), hard negative samples (Wei et al., 2025), and multi-task learning (Tschannen et al., 2025), further improving its effectiveness. The development of large multimodal models further equips embedding models with the ability to follow instructions, cross-modality combinations, and deep understanding and reasoning (Zhang et al., 2025; Jiang et al., 2025c; Li et al., 2026; Lan et al., 2025). However, representing object features and assessing their localization quality are still challenging for global image embedding models. FG-CLIP (Xie et al., 2025b;a) tries to mitigate the problem by introducing regional contrastive learning. It cannot assess localization quality, thus it can only tackle the classification problem. CLARE (Hao et al., 2025) is the first method tackling the object retrieval task by training a traditional detector with contrastive language-instance alignment. However, due to limited data and model capacity, the generalization ability is constrained.

### 2.2. Open-Vocabulary Object Detection

Open-vocabulary object detection aims to detect arbitrary objects described by text queries. Thus, detectors should

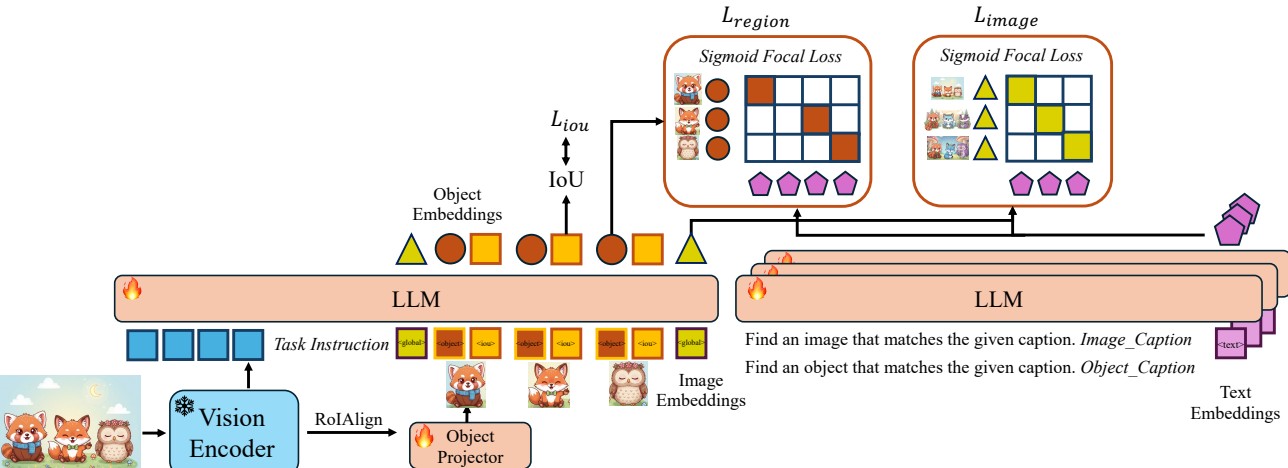

*Figure 2.* The architecture of ObjEmbed. ObjEmbed is a single-tower model built upon a large multimodal language model, enhanced with an object projector and five special tokens (⟨object⟩, ⟨iou⟩, ⟨global⟩, ⟨local_text⟩, and ⟨global_text⟩) whose hidden states from the last layer are used as embeddings. ObjEmbed encodes all object embeddings, IoU embeddings, and the global image embeddings in a single forward pass. The visual embeddings and the text embeddings share the same LLM encoder. For detected objects, object embeddings are initialized from RoI features. And the final matching score is computed as the product of the predicted IoU score (predicted from IoU embeddings) and the classification score (predicted from object embeddings and local text embeddings).

encode objects into embeddings and align them with text embeddings. To align with the text space, some methods (Fu et al., 2025b; Gu et al., 2022; Wu et al., 2023) distill the features of CLIP (Radford et al., 2021), some methods (Fu et al., 2025a; 2024; Du et al., 2024) integrate the CLIP model as a module or backbone, while others (Liu et al., 2024c; Fu et al., 2025c) use deep fusion layers for cross-modal alignment. Although open-vocabulary object detectors excel at object localization, detectors aligning with CLIP cannot truly inherit open-vocabulary capacity, while deep-fusion methods cannot produce query-agnostic object embeddings and the training data is hard to scale up. These limitations motivate us to develop a novel MLLM-based object embedding model with localization awareness and robust transferability.

## 3. Method

### 3.1. Model Architecture

In this work, we aim to build a novel MLLM-based object embedding model, featuring three key properties: (1) **Object-Oriented Representation**: it captures both semantic and spatial aspects of objects and assigns higher matching scores to objects with tighter and more accurate boxes; (2) **Versatility**: the model has the ability to tackle both object-level and image-level tasks; (3) **Efficient Encoding**: the model encodes all objects in an image in a single forward pass. To achieve the goal, we finetune a standard large multimodal model, Qwen3-VL-instruct (Bai et al., 2025a), and introduce five special tokens whose hidden states from the last layer are used as embeddings: (1) ⟨object⟩: the ob-

ject embedding to represent semantic details; (2) ⟨iou⟩: the IoU embedding to assess the box quality of each object; (3) ⟨global⟩: the global image embedding representing the full image; (4) ⟨local_text⟩: the text embedding for matching objects; and (5) ⟨global_text⟩: the text embedding for matching images. The overall architecture is shown in Figure 2 and the different usages of each token are detailed as follows:

**Representing objects as a sequence of embeddings.** Following WeDetect-Ref (Fu et al., 2025a), we first use a universal proposal generator, WeDetect-Uni (Fu et al., 2025a), to generate top-N proposals (100 proposals in this work) for each image. Each RoI feature is extracted via RoIAlign and then compressed into a single token via an object projector. These object features will replace the ⟨object⟩ tokens and are organized sequentially before sending to the large language model. Each object will be separated by prompts "Object i: ⟨object⟩" to ensure distinctiveness. By representing objects as a sequence of embeddings, the model can encode all objects simultaneously with high efficiency.

**Assessing box quality via IoU embeddings.** To equip the model with location awareness, we explicitly model object location by predicting an IoU score for each detected object. We empirically find that using a single token to jointly learn location and classification leads to optimization conflicts. To mitigate this, we introduce a dedicated special token ⟨iou⟩ to represent the quality of each bounding box. This token immediately follows the corresponding object token, forming the structured sequence: "Object i: ⟨object⟩⟨iou⟩." The final IoU score is computed by applying a linear head to the IoU embedding and then multiplying it by the classification score to produce the final object matching score.

**Integrating global image embeddings with object embeddings.** We also introduce a global image token $\langle \text{global} \rangle$ to encode full image content. As demonstrated in the next subsection, we will use both a long text caption and a short text caption as labels to learn global image embeddings. We use two same but separate global tokens for different kinds of captions. The global image embeddings and object embeddings will be encoded in a single forward pass and the final template is:

---
$\langle |\text{vision\_start}| \rangle$ IMAGE $\langle |\text{vision\_end}| \rangle$
Task Instruction
The coarse global image is $\langle \text{global} \rangle$.
Object 0: $\langle \text{object} \rangle \langle \text{iou} \rangle$. $\cdots$ Object N: $\langle \text{object} \rangle \langle \text{iou} \rangle$.
The detailed global image is $\langle \text{global} \rangle$.

---

where IMAGE is the full image. Task instructions are used to separate different tasks, like object detection and REC, and are detailed in Section A.

**Representing text queries as embeddings.** Finally, we introduce a local text token $\langle \text{local\_text} \rangle$ and a global text token $\langle \text{global\_text} \rangle$ to represent text queries. The local text token is used to match object embeddings while the global text token is used to match global image embeddings. The encoding templates are "Find an object that matches the given caption. *CAPTION* $\langle \text{local\_text} \rangle$" and "Find an image that matches the given caption. *CAPTION* $\langle \text{global\_text} \rangle$". We use the same model to encode text and visual embeddings.

**Efficiency Analysis.** With the template described above, all objects in an image, as well as the full image, are encoded in a single forward pass without the need for time-consuming autoregressive token prediction. Each object consumes only 8 tokens. When the full image is encoded into 1000 tokens, the total sequence length remains under 2000, requiring minimal GPU memory and enabling efficient acceleration with FlashAttention-2 (Dao, 2024).

### 3.2. Training Objective

For each image, we annotate a long image caption $C_{long}$, a short image caption $C_{short}$, and a set of objects $\{O_i\}_{i=1}^{M}$, where each object $O_i$ is associated with an object description $C_{obj}^i$ (class names or REC-like descriptions) and a bounding box $B^i$. The overall training objective comprises three components: region-level contrastive learning, image-level contrastive learning, and IoU regression, detailed as follows:

**Region-level contrastive learning.** Unlike traditional contrastive language-image pre-training, where image-caption pairs are one-to-one matched, an object description may correspond to multiple instances, while some proposals remain unmatched due to missing annotations or low-quality bounding boxes. To handle this partial and many-to-one matching, we employ sigmoid focal loss (Lin et al., 2017) for supervi-

sion, a choice commonly adopted in object detection. Specifically, for each object description $C_{obj}^i$, we treat each region proposal $p_j$ as a positive sample if $\text{IoU}(p_j, B^i) > 0.5$, and negative otherwise. We compute the similarity between the proposal object embedding $e_p^j$ and the local text embedding $e_{lt}^i$ of $C_{obj}^i$ as:

$$s_{ij} = \frac{e_{lt}^i \cdot e_p^j}{\|e_{lt}^i\| \|e_p^j\|}. \tag{1}$$

Then, the similarities are optimized via sigmoid focal loss:

$$S_{i,j} = \text{Sigmoid}(\beta_1 \cdot s_{ij} + \mu_1), \tag{2}$$

$$\mathcal{L}_{\text{focal}}(S_{i,j}, y_{i,j})$$
$$= \begin{cases} -\alpha(1 - S_{i,j})^\gamma \log(S_{i,j}), & \text{if } y_{ij} = 1, \\ -(1 - \alpha)(S_{i,j})^\gamma \log(1 - S_{i,j}), & \text{if } y_{ij} = 0, \end{cases} \tag{3}$$

$$\mathcal{L}_{\text{region}} = \sum_{i=1}^{M} \sum_{j=1}^{N} \mathcal{L}_{\text{focal}}(S_{i,j}, y_{ij}), \tag{4}$$

where $y_{ij} = 1$ if $\text{IoU}(p_j, B^i) > 0.5$, and 0 otherwise. $M$ is the number of annotations and $N$ is the number of proposals. $\beta_1$ and $\mu_1$ are learnable parameters. $\alpha$ is set to 0.25 and $\gamma$ is set to 2 following common practice.

**Image-level contrastive learning.** In line with region-level contrastive learning, we also use sigmoid focal loss (Zhai et al., 2023) for image-level supervision $\mathcal{L}_{\text{image}}$. In this setting, each image is paired with a single caption in a one-to-one manner. Given the global image embedding $e_g^j$ and the associated global text embedding $e_{gt}^i$ from either $C_{long}$ or $C_{short}$, the image-level loss $\mathcal{L}_{\text{image}}$ is defined as:

$$\hat{S}_{i,j} = \text{Sigmoid}(\beta_2 \cdot \frac{e_{gt}^i \cdot e_g^j}{\|e_{gt}^i\| \|e_g^j\|} + \mu_2), \tag{5}$$

$$\mathcal{L}_{\text{image}} = \sum_{i=1}^{K} \sum_{j=1}^{K} \mathcal{L}_{\text{focal}}(\hat{S}_{i,j}, \hat{y}_{ij}), \tag{6}$$

where $\hat{y}_{ij} = 1$ if $i = j$, and 0 otherwise. We collect captions from other GPUs to enlarge the negative samples. Here, $K$ is the global batch size, and $\beta_2$ and $\mu_2$ are learnable parameters that are not shared with those of the object embeddings. Additionally, the first global image embedding is only supervised by short captions while the second global image embedding is only supervised by long captions. The corresponding losses are computed separately and then combined in the overall objective.

**IoU regression.** The IoU loss $\mathcal{L}_{\text{iou}}$ is also formulated as sigmoid focal loss with ground truth IoUs $u_j^*$ between proposals and corresponding boxes as labels:

$$\mathcal{L}_{\text{iou}} = -\sum_{j=1}^{N^+} \mathcal{L}_{\text{focal}}(\hat{u}_j, u_j^*), \tag{7}$$

*Table 1.* Object detection results. Specialist detectors are typically designed for target tasks and trained on the target datasets.

| Method | COCO | | | | COCO-O | ODinW13 | FG-OVD | | | | D3 | | |
|---|---|---|---|---|---|---|---|---|---|---|---|---|---|
| | AP | $AP_s$ | $AP_m$ | $AP_l$ | AP | AP | Hard | Medium | Easy | Trivial | Full | Pres | Abs |
| *Specialist Detectors* | | | | | | | | | | | | | |
| DINO-R50 (Zhang et al., 2023) | 51.2 | 35.0 | 54.3 | 65.3 | - | - | - | - | - | - | - | - | - |
| GUIDED (Li et al., 2025a) | - | - | - | - | - | - | 57.5 | 69.5 | 73.3 | 72.6 | - | - | - |
| Weak-to-Strong (Park et al., 2024) | - | - | - | - | - | - | - | - | - | - | 30.8 | 31.0 | 30.4 |
| *Open-vocabulary Detectors* | | | | | | | | | | | | | |
| GLIP-T (Li et al., 2022b) | 46.1 | - | - | - | 29.0 | 46.5 | - | - | - | - | 19.1 | 18.3 | 21.5 |
| OWLv2 (L/14) (Minderer et al., 2023) | 37.9 | 24.9 | 41.2 | 53.7 | 42.7 | 50.1 | 25.4 | 41.2 | 42.8 | 63.2 | 22.8 | 22.1 | 24.7 |
| Grounding-DINO-T (Liu et al., 2024c) | 47.9 | 33.4 | 51.2 | 62.2 | 37.6 | 51.4 | 17.0 | 28.4 | 31.0 | 62.5 | 20.7 | 20.1 | 22.5 |
| LLMDet-T (Fu et al., 2025c) | **54.9** | **40.1** | 58.3 | 68.6 | 36.1 | 52.1 | 15.0 | 26.2 | 23.8 | 55.4 | 17.2 | 17.1 | 17.6 |
| WeDetect-B (Fu et al., 2025a) | 52.1 | 34.8 | 57.1 | 69.2 | 44.1 | **53.1** | - | - | - | - | - | - | - |
| *MLLMs* | | | | | | | | | | | | | |
| VLM-FO1-3B (Liu et al., 2025c) | 44.0 | - | - | - | - | 44.0 | - | - | - | - | - | - | - |
| LMM-Det-7B (Li et al., 2025b) | 47.5 | 34.7 | 51.8 | 60.3 | - | - | - | - | - | - | - | - | - |
| WeDetect-Ref-2B (Fu et al., 2025a) | 49.9 | 34.0 | 58.0 | 68.9 | 55.8 | 48.2 | 28.7 | 42.5 | 48.1 | 65.6 | 41.8 | 43.9 | 35.4 |
| WeDetect-Ref-4B (Fu et al., 2025a) | 50.0 | 34.7 | 57.6 | 69.2 | 56.0 | 47.3 | 31.1 | 45.7 | 50.1 | 68.4 | **42.0** | **44.0** | **35.8** |
| *Ours* | | | | | | | | | | | | | |
| Qwen3-VL-2B (Bai et al., 2025a) | 16.9 | 6.2 | 20.7 | 36.4 | 29.4 | 43.4 | 59.3 | 58.9 | 53.8 | 55.9 | 16.0 | 17.0 | 11.2 |
| Qwen3-VL-4B (Bai et al., 2025a) | 29.1 | 12.9 | 33.9 | 54.2 | 44.4 | 48.2 | 62.0 | 62.6 | 64.0 | 45.7 | 24.3 | 25.8 | 19.7 |
| ObjEmbed-2B | 52.9 | 35.8 | **59.7** | **72.5** | **59.1** | 49.8 | 65.7 | **74.3** | **77.3** | **76.2** | 39.4 | 41.1 | 34.2 |
| ObjEmbed-4B | 53.0 | 35.6 | 59.6 | 72.2 | 58.0 | 50.8 | **66.1** | 74.1 | 77.2 | 76.0 | 40.2 | 42.2 | 34.4 |

*Table 2.* The overview of ObjEmbed training dataset.

| Type | #Sample | Dataset |
|---|---|---|
| DET | 380k | COCO (111k) (Lin et al., 2014), LVIS (94k) (Gupta et al., 2019) V3Det (175k) (Wang et al., 2023) |
| REC | 909k | RefCOCO/+/g (28k) (Kazemzadeh et al., 2014; Yu et al., 2016; Mao et al., 2016) FG-OVD (175k) (Bianchi et al., 2024), HumanRef (43k) (Jiang et al., 2025b) grefcoco (15k) (Liu et al., 2023), Ref-L4 (9k) (Chen et al., 2025) DAM (77k) (Lian et al., 2025), FineCops-Ref (29k) (Liu et al., 2024b) REIRCOCO (25k) (Hao et al., 2025), self-collected data (508k) |
| SUM | 1289k | |

where $\hat{u}_j$ is predicted IoU scores. Since not all objects in an image are annotated, $\mathcal{L}_{iou}$ is only applied to $N^+$ positive proposals.

The total training objective is a weighted combination:

$$\mathcal{L}_{total} = \lambda_1 \mathcal{L}_{region} + \lambda_2 \mathcal{L}_{image} + \lambda_3 \mathcal{L}_{iou}, \qquad (8)$$

where $\lambda_1$, $\lambda_2$, and $\lambda_3$ are hyperparameters.

### 3.3. Training Data Construction

To reduce annotation costs, we aggregate existing open-source object detection and referring expression comprehension datasets that provide region-level annotations. Due to limited data coverage, we further curate 300k images from SA-1B (Kirillov et al., 2023) and 200k images self-crawled from licensed websites. These images are first annotated with class-agnostic bounding boxes using WeDetect-Uni (Fu et al., 2025a), followed by generating unique instance-level descriptions for each object using Qwen3-VL-235B (Bai et al., 2025a). We find that the instance-level descriptions should be as unique as possible to minimize false-negative conflicts and can not include subjective content. Finally, each image is assigned both a short and a long caption, also generated by Qwen3-VL-235B. The prompts used for annotation are provided in Section B.

The overall training data are summarized in Table 2, comprising 1.3M images and 8.1M bounding boxes.

## 4. Experiment

### 4.1. Implementation Details

ObjEmbed is finetuned from Qwen3-VL-Instruct (Bai et al., 2025a) by first initializing the object projector following WeDetect-Ref (Fu et al., 2025a) and then training under the objective described in Equation (8). The model is trained using 16 GPUs, with a batch size of 2 images per GPU, and an initial learning rate of $2e^{-5}$. All parameters are updated during training except those in the frozen vision encoder. Training proceeds for two epochs. The loss weights $\lambda_1$, $\lambda_2$, and $\lambda_3$ are set to 1.0, 1.0 and 0.25. Input images are resized such that the number of visual tokens ranges from 900 to 1200, corresponding to 900*32*32 to 1200*32*32 pixels, ensuring adaptive computation based on image content.

### 4.2. Comparisons on Fine-Grained Region-Level Tasks

**Results on object detection benchmarks.** Object detection aims to simultaneously localize and classify all target objects within an image. The standard evaluation metric is mAP, computed over IoU thresholds ranging from 0.50 to 0.95, which places strong emphasis on localization accuracy. We evaluate on five benchmarks: COCO (Lin et al., 2014) for general object detection, COCO-O (Mao et al., 2023) for out-of-distribution detection, ODinW13 (Li et al., 2022a) for detection in the wild, FG-OVD (Bianchi et al., 2024) for fine-grained attribute-aware recognition, and D3 (Xie et al., 2023) for language-based object detection. For COCO, COCO-O, and ODinW13, the text queries are typically category-level labels, such as "person". Each image in these datasets contains multiple instances of target objects. As a result, accurate localization of individual objects is crucial. In contrast, the text queries in FG-OVD and D3 are significantly more detailed and descriptive, for example, "A dark green bench with a brown wooden back and seat and metal

*Table 3.* Evaluation results on referring expression comprehension datasets. The evaluation metric is the Top-1 accuracy.

| Method | RefCOCO | | | RefCOCO+ | | | RefCOCOg | | Avg. |
|--------|---------|---------|---------|----------|---------|---------|----------|------|------|
| | val | testA | testB | val | testA | testB | val | test | |
| Grounding-DINO-L (Liu et al., 2024c) | 90.6 | 93.2 | 88.2 | 82.8 | 89.0 | 75.9 | 86.1 | 87.0 | 86.6 |
| CLARE-L (Hao et al., 2025) | 91.4 | 92.0 | 90.0 | 80.1 | 83.3 | 74.4 | 86.3 | 86.7 | 85.5 |
| Qwen2.5-VL 3B (Bai et al., 2025b) | 89.1 | 91.7 | 84.0 | 82.4 | 88.0 | 74.1 | 85.2 | 85.7 | 85.0 |
| Qwen2.5-VL 7B (Bai et al., 2025b) | 90.0 | 92.5 | 85.4 | 84.2 | 89.1 | 76.9 | 87.2 | 87.2 | 86.6 |
| InternVL2.5-8B (Chen et al., 2024b) | 90.3 | **94.5** | 85.9 | 85.2 | 91.5 | 78.8 | 86.7 | 87.6 | 87.6 |
| InternVL3.5-38B (Wang et al., 2025) | 90.3 | 91.8 | 89.0 | 87.5 | 90.0 | **84.7** | 89.7 | 89.9 | 89.1 |
| Octopus 7B (Zhao et al., 2024) | 89.0 | 92.6 | 83.4 | 83.6 | 89.4 | 76.0 | 84.3 | 86.3 | 85.6 |
| VLM-R1 3B (Shen et al., 2025) | 90.1 | 92.3 | 85.2 | 84.2 | 89.4 | 76.8 | 85.6 | 86.8 | 86.3 |
| Rex-Omni 3B (Jiang et al., 2025a) | 86.6 | 89.5 | 82.8 | 79.6 | 84.8 | 71.4 | 85.3 | 86.2 | 83.3 |
| ChatRex 7B (Jiang et al., 2024) | 91.0 | 94.1 | 87.0 | **89.8** | 91.9 | 79.3 | 89.8 | 90.0 | 89.1 |
| VLM-FO1 3B (Liu et al., 2025b) | 91.1 | 93.7 | 87.6 | 86.4 | 91.9 | 80.6 | 88.9 | 88.3 | 88.6 |
| RexSeek 7B (Jiang et al., 2025b) | - | - | - | - | - | - | 84.0 | 84.4 | - |
| Qwen3-VL 2B (Bai et al., 2025a) | 88.2 | 91.0 | 83.1 | 78.6 | 85.2 | 70.4 | 84.7 | 85.0 | 83.3 |
| Qwen3-VL 4B (Bai et al., 2025a) | 90.7 | 92.2 | 86.7 | 82.9 | 89.4 | 75.6 | 87.3 | 87.7 | 86.6 |
| ObjEmbed-2B | 91.7 | 93.5 | 88.3 | 83.4 | 90.1 | 76.6 | 88.6 | 88.6 | 87.6 |
| ObjEmbed-4B | **92.5** | 94.3 | **90.2** | 86.1 | 91.4 | 80.6 | **89.9** | **90.7** | **89.5** |

*Table 4.* Evaluation results on local image retrieval datasets. The evaluation metric for SORCE-1K and REIRCOCO is Recall@1 while the evaluation metric for ILIAS is mAP@50, a metric evaluating the top 50 predictions.

| Method | SORCE-1K | REIRCOCO | ILIAS | | Avg. |
|--------|----------|----------|-------|-----|------|
| | T2I | T2I | T2I | I2I | |
| CLIP ViT-L/14 | 32.6 | 16.0 | 44.2 | 39.9 | 33.2 |
| SigLIP2 ViT-So/16 | 34.2 | 22.9 | 45.2 | 55.4 | 39.4 |
| MetaCLIP2 ViT-H/14 | 37.5 | 19.2 | 49.6 | 42.6 | 37.2 |
| FG-CLIP2 ViT-So/16 | 46.9 | 27.3 | 57.9 | 64.1 | 49.1 |
| GME-2B | 28.3 | 19.8 | 35.7 | 44.9 | 32.2 |
| GME-7B | 30.6 | 24.6 | 33.5 | 40.4 | 32.3 |
| VLM2Vec-2B | 26.3 | 20.1 | 29.8 | 39.7 | 29.0 |
| VLM2Vec-V2-2B | 24.8 | 24.5 | 32.0 | 36.4 | 29.4 |
| QQMM-embed-v2-7B | 47.7 | 32.7 | 44.1 | 48.0 | 43.1 |
| RzenEmbed-7B | 36.0 | 30.4 | 39.2 | 42.0 | 36.9 |
| UME-R1-2B | 36.1 | 23.3 | 31.6 | 38.9 | 32.5 |
| UME-R1-7B | 41.3 | 28.2 | 37.6 | 41.7 | 37.2 |
| Qwen3-VL-Embedding-2B | 45.4 | 28.0 | 47.3 | 57.9 | 44.7 |
| Qwen3-VL-Embedding-8B | 49.1 | 32.6 | 51.2 | 59.2 | 48.0 |
| FG-CLIP2 ViT-So/16 (RoIAlign) | 28.4 | 16.0 | 53.9 | 72.5 | 42.7 |
| ObjEmbed-2B | 67.3 | 37.5 | 76.5 | 84.0 | 66.3 |
| ObjEmbed-4B | **71.7** | **39.3** | **77.6** | **85.3** | **68.5** |

arms and legs". These fine-grained textual descriptions require models to precisely align multiple attributes, such as color, material, and structure, with the corresponding regions in the image, posing a greater challenge for accurate vision-language grounding.

As shown in Table 1, traditional detectors excel at precise localization and handling multi-object scenes, but their semantic understanding is limited to fixed class vocabularies with short category names. Consequently, their performance on FG-OVD and D3 is low. In contrast, multimodal large language models (MLLMs) demonstrate superior language comprehension and can interpret complex textual descriptions, yet suffer from poor localization accuracy due to coarse spatial reasoning, resulting in low performance on COCO and ODinW13. Equipped with the dual-token design, ObjEmbed achieves a favorable balance between high semantic discriminability and precise location assessment. It understands rich language inputs, prioritizes well-localized predictions with higher scores, and is robust to domain variances, leading to consistently strong performance across all five benchmarks.

**Results on referring expression comprehension benchmarks.** Referring expression comprehension aims to localize the unique object in an image that is described by a natural language expression. This task demands fine-grained multimodal reasoning, including deep linguistic understanding, interpretation of complex phrases, and contextual reasoning over spatial and semantic relationships among objects. The standard evaluation metric is accuracy at an IoU threshold of 0.5. We evaluate on three standard benchmarks: RefCOCO (Kazemzadeh et al., 2014), RefCOCO+ (Yu et al., 2016), and RefCOCOg (Mao et al., 2016). As shown in Table 3, ObjEmbed achieves an average accuracy of 89.5, surpassing both specialized MLLMs designed for referring tasks and significantly larger general-purpose multimodal models. This strong performance demonstrates the high semantic discriminability of our object embeddings.

**Results on local image retrieval benchmarks.** Local image retrieval aims to retrieve a target image from a gallery based on a query that describes only a small region or specific object within it. The query can be a textual description (text-to-image, T2I) or an image exemplar (image-to-image, I2I), both requiring fine-grained cross-modal alignment between local regions and external queries. For text-based retrieval (T2I), we evaluate on three benchmarks: SORCE-1K (Liu et al., 2025a), REIRCOCO (Hao et al., 2025), and ILIAS (Kordopatis-Zilos et al., 2025). On REIRCOCO and ILIAS, we adopt a simplified evaluation protocol (detailed in Section C). The evaluation metric is Recall@1 for SORCE-1K and REIRCOCO, and mAP@50 for ILIAS. These metrics assess whether the correct target images are ranked highly in the retrieval results, without considering

*Table 5.* Evaluation results on global image retrieval datasets. The evaluation metrics are Recall@1.

| Method | Long Image Captions | | | | Short Image Captions | | | | Multilingual Image Captions | | | | Avg. |
|---|---|---|---|---|---|---|---|---|---|---|---|---|---|
| | ShareGPT4V | | DCI | | COCO | | Flickr30K | | COCO-CN | | Flickr30K-CN | | |
| | I2T | T2I | I2T | T2I | I2T | T2I | I2T | T2I | I2T | T2I | I2T | T2I | |
| CLIP ViT-L/14 (Radford et al., 2021) | 86.5 | 83.6 | 37.2 | 36.4 | 58.0 | 37.1 | 87.4 | 67.3 | - | - | - | - | - |
| EVA-CLIP ViT-L/14 (Sun et al., 2023) | 91.5 | 89.4 | 47.2 | 47.8 | 64.2 | 47.9 | 89.2 | 77.9 | - | - | - | - | - |
| SigLIP2 ViT-So/16 (Tschannen et al., 2025) | 78.6 | 79.5 | 46.0 | 47.1 | 71.0 | 55.8 | 94.1 | 82.5 | 72.0 | 50.7 | 78.4 | 51.7 | 67.3 |
| MetaCLIP2 ViT-H/14 (Chuang et al., 2025) | 93.9 | 89.2 | 53.0 | 50.2 | 66.8 | 47.7 | 91.9 | 77.0 | 80.1 | 63.1 | 89.3 | 72.2 | 72.9 |
| FG-CLIP2 ViT-So/16 (Xie et al., 2025a) | 97.5 | 96.7 | 70.6 | 72.1 | 74.6 | 56.7 | 95.9 | 85.0 | 83.2 | 68.1 | 91.5 | 77.2 | 80.8 |
| GME-2B (Zhang et al., 2025) | 92.8 | 91.7 | 52.9 | 58.8 | 67.3 | 51.7 | 88.0 | 75.1 | 79.7 | 66.7 | 85.5 | 71.1 | 73.4 |
| GME-7B (Zhang et al., 2025) | 89.3 | 92.2 | 59.8 | 64.6 | 67.8 | 55.6 | 91.1 | 81.1 | 81.7 | 71.8 | 91.8 | 79.1 | 77.2 |
| VLM2Vec-2B (Jiang et al., 2025c) | 77.7 | 74.3 | 33.2 | 44.2 | 53.5 | 39.2 | 83.0 | 68.5 | 67.9 | 52.6 | 76.1 | 56.9 | 60.6 |
| VLM2Vec-V2-2B (Meng et al., 2025) | 92.1 | 92.4 | 53.1 | 65.3 | 62.6 | 50.3 | 89.2 | 80.3 | 74.4 | 60.3 | 84.0 | 67.1 | 72.6 |
| UME-R1-2B (Lan et al., 2025) | 92.9 | 93.9 | 59.8 | 67.9 | 71.4 | 54.4 | 91.6 | 79.1 | 83.6 | 70.1 | 87.9 | 75.0 | 77.3 |
| Qwen3-VL-Embedding-2B (Li et al., 2026) | 97.8 | 96.7 | 77.9 | 79.7 | 69.6 | 55.2 | 92.9 | 81.9 | 83.0 | 72.2 | 91.9 | 78.6 | 81.5 |
| ObjEmbed-2B | 97.1 | 97.3 | 74.5 | 74.6 | 75.2 | 51.0 | 94.2 | 80.0 | 86.0 | 66.7 | 94.0 | 76.1 | 80.6 |
| ObjEmbed-4B | 97.5 | 97.7 | 77.2 | 76.9 | 75.7 | 52.2 | 94.2 | 80.4 | 87.4 | 68.8 | 94.7 | 77.3 | 81.7 |

bounding box localization accuracy. In our framework, we compute the overall image similarity by taking the maximum matching score among all objects. As shown in Table 4, we compare ObjEmbed with other global embedding models that represent the entire image as a single, holistic feature vector. However, such global representations suffer from semantic ambiguity and fail to capture fine-grained details, leading to poor performance in local retrieval tasks. Even FG-CLIP2 (Xie et al., 2025a), which incorporates regional contrastive learning, underperforms significantly in T2I retrieval when using object embeddings extracted by RoIAlign with the same object proposals and scoring strategy as ours. We hypothesize that this is due to misalignment between region features and complex queries. In contrast, our model represents each object with fine-grained details and gets an average score of 68.5, surpassing other models by around 20 points. For the image-based retrieval (I2I) task, we use global image embeddings to represent image exemplars and to retrieve target images. Surprisingly, although our model does not optimize for image-based retrieval tasks, the model aligns global text embeddings and global image embeddings in a unified semantic space. Therefore, it can transfer from text-based retrieval tasks to image-based retrieval tasks successfully.

### 4.3. Comparisons on Image-Level Tasks

In Table 5, we evaluate ObjEmbed on traditional image-text retrieval benchmarks, including long text retrieval (ShareGPT4V (Chen et al., 2024a) and DCI (Urbanek et al., 2024)), short text retrieval (COCO (Chen et al., 2015) and Flickr30K (Young et al., 2014)), and multilingual text retrieval (COCO-CN (Li et al., 2019) and Flikr30K-CN (Lan et al., 2017)). Despite using a relatively small-scale training set, ObjEmbed achieves an overall score of 81.7 points, which is highly competitive with both traditional CLIP-style models and recent LMM-based embedding approaches. Moreover, it demonstrates robust generalization across varying text lengths and multiple languages.

*Table 6.* Ablation studies on different object token designs.

| Method | COCO | | | | RefCOCO |
|---|---|---|---|---|---|
| | AP | $AP_s$ | $AP_m$ | $AP_l$ | Avg. |
| single token & label=1 | 37.1 | 28.8 | 47.9 | 52.8 | 86.8 |
| single token & label=IoU | 42.3 | 29.5 | 50.9 | 60.9 | 87.1 |
| two tokens (iou+cls) | 45.1 | 27.1 | 51.1 | 67.1 | 86.8 |
| two tokens (cls+iou) | 45.5 | 27.3 | 51.4 | 66.6 | 86.6 |

*Table 7.* Ablation studies on instructions.

| Method | COCO | | | | RefCOCO |
|---|---|---|---|---|---|
| | AP | $AP_s$ | $AP_m$ | $AP_l$ | Avg. |
| None | 42.1 | 28.0 | 52.5 | 63.1 | 86.9 |
| object instruction | 45.5 | 27.3 | 51.4 | 66.6 | 86.6 |
| object & task instructions | 47.1 | 32.1 | 55.6 | 67.1 | 86.9 |

### 4.4. Ablation Study

In this subsection, we explore the effects of different designs with the 4B model, partial data, and a one-epoch training schedule.

**Ablation studies on object token design.** As objects are sensitive to localization quality, an object embedding model should have the ability to assess the quality of boxes. To achieve the goal, we change classification labels in sigmoid focal loss to box IoUs, which increases 5.2% mAP on COCO, as shown in Table 6. However, the classification and box localization may have conflicting training objectives. We find that decoupling an object into two tokens, one for classification and the other for IoU regression, further improves 3.2% mAP on COCO. And placing the classification token in front of the IoU token is slightly better. Since REC places less emphasis on precise localization, its performance is relatively robust to the choice of IoU design.

**Ablation studies on instructions.** As shown in Table 7, we study the effects of instructions. As we encode multiple objects simultaneously, using the object instruction ("Object i: ⟨object⟩⟨iou⟩.") to separate different objects can increase the distinctiveness of each object and increase 3.4% mAP

*Table 8.* Ablation studies on different training objectives. LIR* denotes the average scores of local image retrieval tasks except for the I2I task. GIR denotes the average scores of global image retrieval tasks.

| Method | COCO | RefCOCO | LIR* | GIR |
|---|---|---|---|---|
| object-level | 53.0 | 88.1 | 60.2 | - |
| image-level | - | - | - | 81.2 |
| object-level & image-level | 52.8 | 87.4 | 62.9 | 81.6 |

*Table 9.* Ablation studies on image-level supervision design. 'Share' denotes whether global text embeddings and local text embeddings share the same special token. '#token' denotes the number of global image tokens and 'Type' can be a single long caption (long), a single short caption (short), a single randomly selected caption (mix), or two captions used simultaneously (both). 'LIR' denotes the average scores of local image retrieval tasks. 'GIR' denotes the average scores of global image retrieval tasks.

| Exp. | Share | #token | Type | COCO | RefCOCO | LIR | GIR |
|---|---|---|---|---|---|---|---|
| 1 | ✓ | 1 | mix | 52.4 | 86.0 | 67.3 | 80.1 |
| 2 | | 1 | mix | 52.7 | 86.8 | 67.6 | 80.1 |
| 3 | | 1 | long | 52.6 | 86.1 | 67.2 | 72.7 |
| 4 | | 1 | short | 52.6 | 86.3 | 67.4 | 80.2 |
| 5 | | 1 | both | 52.8 | 86.6 | 67.5 | 80.6 |
| 6 | | 2 | both | 52.8 | 87.4 | 68.6 | 81.6 |

on COCO. Further, different tasks focus on different aspects of objects. Object detection focuses on the shared properties within a class while referring expression comprehension requires encoding instance-specific features. Incorporating task instructions to guide the model encoding different features for different tasks is beneficial. Task prompts used in this work are shown in Section A.

**Ablation studies on different training objectives.** In this work, we build a universal object embedding model along with the global image representation ability. In Table 8, we find that incorporating an image-level training objective with an object-level training objective can increase the local image retrieval performance by 2.7 points while maintaining the performance on object detection and REC. Further, comparing with single image-level training, training with both objectives can also boost the global image retrieval performance by 0.4 points, demonstrating the mutual benefits between the two training objectives.

**Ablation studies on image-level supervision design.** In this work, we use both local and global text embeddings and long and short captions as global text embeddings to learn global image embeddings. We study their effects in Table 9. As local text embeddings match with object embeddings while global text embeddings match with global image embeddings, we find that not sharing the text tokens can mitigate task discrepancies and get higher performance on COCO (+0.3% mAP) and RefCOCO (+0.8), comparing Exp1 and Exp2. Further, using only short captions, long captions, or a mixture of them can not handle complex retrieval requirements. Using both captions for supervision achieves

*Table 10.* Effect of the proposal network.

| Exp. | proposal network | #proposals | COCO AR/AP | RefCOCO AR$_{50}$/Top1 | LIR | GIR |
|---|---|---|---|---|---|---|
| 1 | WeDetect-Uni (Fu et al., 2025a) | 50 | 62.9/52.6 | 98.4/89.01 | 67.4 | 81.7 |
| 2 | WeDetect-Uni (Fu et al., 2025a) | 100 | 66.7/53.0 | 99.2/89.46 | 68.5 | 81.7 |
| 3 | WeDetect-Uni (Fu et al., 2025a) | 150 | 68.0/51.9 | 99.4/89.51 | 68.3 | 81.7 |
| 4 | UPN (Jiang et al., 2024) | 100 | 69.7/53.0 | 98.8/89.45 | 66.7 | 81.7 |

*Table 11.* Effect of the quality of proposals. 'mix' denotes a setting where a portion of the generated object proposals are randomly replaced with ground truth bounding boxes. We report fixed AP (Dave et al., 2021) on LVIS. 'AR' is computed as the mean recall across IoU thresholds ranging from 0.50 to 0.95.

| mix | COCO | | | | | LVIS v1 val | | | | |
|---|---|---|---|---|---|---|---|---|---|---|
| | AR | AP | AP$_s$ | AP$_m$ | AP$_l$ | AR | AP | AP$_s$ | AP$_m$ | AP$_l$ |
| | 66.7 | 53.0 | 35.6 | 59.6 | 72.2 | 50.8 | 49.0 | 53.7 | 50.2 | 45.5 |
| ✓ | 100.0 | 65.2 | 55.3 | 70.1 | 77.5 | 100.0 | 66.6 | 66.7 | 67.1 | 65.9 |

the highest 80.6 global retrieval results (Exp2-Exp5). Finally, using two global image tokens (Exp6), one for short captions and one for long captions, gets the highest results. And we find that the performance on object detection and REC is quite robust to the image-level designs.

### 4.5. Discussion

**Is ObjEmbed robust to the proposal network?** In ObjEmbed, we employ a state-of-the-art proposal generator, WeDetect-Uni (Fu et al., 2025a), to produce 100 object proposals per image. We find that ObjEmbed exhibits strong robustness to the choice of proposal network. As shown in Table 10, ObjEmbed achieves similar performance across different numbers of proposals (50, 100, 150) and various proposal architectures (e.g., UPN (Jiang et al., 2024)). The model effectively prioritizes high-quality proposals with higher matching scores, while suppressing low-quality ones. As a result, as long as object recall is comparable, the final performance remains stable. Furthermore, the quality of the proposals does not affect the global retrieval results.

**How does proposal quality affect performance?** In ObjEmbed, the recall rate is essential as the model can not encode missing objects. As shown in Table 11, WeDetect-Uni achieves an Average Recall (AR) of 66.7 on COCO and 50.8 on LVISv1 val (Gupta et al., 2019). To assess the upper bound, we conduct an oracle experiment in which ground truth bounding boxes are randomly mixed into the generated proposals (denoted as 'mix' in the table). With access to ground truth regions, ObjEmbed achieves a significant gain of 12.2% AP on COCO and 17.6% AP on LVIS, demonstrating that (1) the model can accurately align object proposals with corresponding text descriptions, and (2) it effectively ranks high-quality proposals above low-quality ones during matching. These results indicate that the representation learning in ObjEmbed is orthogonal to proposal quality but its overall performance is still limited by the recall rate of proposals. Therefore, improving proposal quality through

*Table 12.* Ablation studies on supporting box regression.

| box regression | COCO | RefCOCO | LIR | GIR |
|:---:|:---:|:---:|:---:|:---:|
| ✓ | 52.5 | 86.2 | 68.1 | 81.5 |
|  | 52.8 | 87.4 | 68.6 | 81.6 |

fine-tuning the proposal network on target datasets or leveraging human-annotated bounding boxes can further boost performance.

**Can we directly use ObjEmbed to regress high-quality bounding boxes, similar to object detectors?** Given the importance of proposal quality, a natural question arises: can the embedding model itself be leveraged to refine proposals by predicting bounding box offsets, thereby improving localization accuracy? To investigate this, we conduct experiments where the model uses the IoU embeddings to predict both IoU scores and bounding box offsets simultaneously. The offset regression head is trained with a combination of L1 loss and IoU loss, following the standard practice in DETR (Carion et al., 2020). Similar to IoU regression loss, the bounding box regression loss is applied only to positive proposals with an IoU greater than 0.5. As a result, the regression head can refine existing bounding boxes but is unable to generate missing ones. As shown in Table 12, we find that incorporating box regression degrades overall performance, possibly due to learning conflicts.

### 4.6. Efficiency Analysis

ObjEmbed enjoys superior efficiency across multiple tasks.

For retrieval tasks, ObjEmbed generates both object-level and global image embeddings in a single forward pass, similar to conventional global embedding models. Under this setting, ObjEmbed-4B achieves an inference speed of 6.9 fps, which is comparable to Qwen3-VL-Embedding-8B running at 9.5 fps.

For referring expression comprehension (REC), ObjEmbed requires two forward passes: one to encode object embeddings, and another to encode the text query. The final alignment is computed efficiently via dot-product similarity. This design enables ObjEmbed-4B to achieve 4.0 fps. In contrast, the base model Qwen3-VL-4B must autoregressively decode object coordinates token by token, resulting in significantly slower inference at only 0.4 fps.

For object detection, the class text embeddings can be precomputed and shared across all images. ObjEmbed thus only requires a single forward pass to compute object embeddings per image, allowing ObjEmbed-4B to run at 6.9 fps. In comparison, Qwen3-VL-4B struggles with efficiency, requiring several minutes per sample due to the sequential decoding of hundreds of tokens for multiple objects.

## 5. Limitation

As a pioneering MLLM-based object embedding model, ObjEmbed can be further improved in the following directions, which are left for future work.

Scaling up training data. Due to resource constraints, we currently train on only 1.3M samples, significantly fewer than those used in CLIP-series models. Scaling up the pre-training data through broader data collection could enhance model performance.

Hard negative mining. Hard negatives are essential for learning discriminative embeddings. However, effective mining must be balanced with the mitigation of the false negative problem. Integrating robust hard negative sampling strategies while accounting for annotation incompleteness can further boost performance.

## 6. Conclusion

In this work, we present ObjEmbed, a novel MLLM-based object embedding model that features object-oriented representation, versatility, and efficient encoding. In our framework, each object is represented by two complementary embeddings: an object embedding for semantic matching and an IoU embedding for assessing localization quality. This decoupled design reduces learning complexity while maintaining encoding efficiency. ObjEmbed can be seamlessly applied to a wide range of downstream tasks, including object detection, referring expression comprehension, local image retrieval, and global image retrieval. The consistently high and balanced performance across 18 diverse benchmarks demonstrates the effectiveness and generalization capability of our approach.

## Acknowledgements

This work was supported partially by National Key Research and Development Program of China (2024YFA1011900), NSFC (92470202), Guangdong NSF Project (No. 2023B1515040025), Guangdong Key Research and Development Program (No.2024B0101040004, No. 2025B0909020002), Project of Guangdong Provincial Key Laboratory of Information Security Technology (2023B1212060026), Key Areas Research plan of the Guangdong S&T programme (2025B0101120008).

## Impact Statement

This paper presents work whose goal is to advance the field of Machine Learning. There are many potential societal consequences of our work, none which we feel must be specifically highlighted here.

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

# A. Details of Task Instructions

ObjEmbed is a versatile embedding model applicable to a wide range of downstream tasks. However, different tasks emphasize distinct aspects of object representation. For instance, object detection relies on shared semantic properties within object categories, whereas referring expression comprehension requires fine-grained, instance-specific features to distinguish between visually similar objects.

To mitigate potential task conflicts, we introduce task-specific instructions to guide the model in generating context-aware embeddings tailored to each task. The instructions used during training are listed as follows:

**Object Detection**

- Detect all objects in the image by identifying the common visual features of their respective classes.

- Localize each object by matching it to the archetypal visual form of its category.

- Detect all objects in the image by recognizing the shared visual attributes of their respective categories.

- Identify every object in the scene based on the core visual characteristics that define its class.

- Locate all objects by using the fundamental visual properties common to their object class.

- Perform object detection by referencing the shared visual patterns that characterize each class.

- Find every object in the picture by matching it to the defining visual traits of its category.

- Identify all objects present by their class-defining visual features, which are common across all instances.

- Detect every object based on the visual essence shared by all members of its class.

- Localize each object in the image according to the general visual blueprint of its category.

- Identify all objects by applying the common visual criteria that define their respective classes.

**Referring expression comprehension**

- Locate the specific object being described by analyzing its unique instance-level attributes, its spatial position, and its relationship with surrounding objects.

- Identify the single instance mentioned in the text by considering its distinct visual features, its location within the scene, and its context relative to nearby items.

- Ground the referring expression by pinpointing the object that matches the description's details regarding its appearance, placement, and interaction with other elements.

- Find the objects that correspond to the given description, paying close attention to its specific details, its position, and how it relates to its neighbors.

- Disambiguate and find the correct object by carefully examining the provided description of its instance-specific properties, its coordinates in the image, and its spatial arrangement with other objects.

- Resolve the reference by identifying the object that uniquely matches the specified details, including its appearance, its place in the scene, and its connections to adjacent objects.

- Pinpoint the described instance by evaluating its specific visual traits, its spatial context, and its relational properties with other objects in the image.

- To locate the referred object, you must analyze three things from the description: 1) its unique visual details (e.g., color, texture), 2) its precise location, and 3) its relationship to the objects around it.

- Find the specific object the text is referring to by synthesizing information about its individual characteristics, its location, and its interactions within the scene.

**Celebrity**

- Identify the famous person depicted in this image.

- Recognize and name the public figure featured in this picture.

- Please provide the name of the well-known individual in this image.

- Identify all recognizable celebrities in this image.

- State the name of the celebrity shown.

## B. Details of Dataset Annotation

To train ObjEmbed, each image in the training dataset is annotated with a long caption, a short caption, and several regions of interest, where each region is associated with a corresponding object description. To minimize false-negative conflicts during contrastive learning, captions are designed to be as diverse and distinctive as possible. Furthermore, they should exclude subjective or interpretive content to ensure objectivity and consistency. To achieve high-quality annotations, we carefully design structured prompts and employ a state-of-the-art multimodal large language model, Qwen-VL-235B (Bai et al., 2025a), as the automated annotator. Since image captioning is a fundamental capability of MLLMs (Lin et al., 2025; Bai et al., 2025a), the generated captions exhibit minimal hallucination. The annotation prompts are as follows:

---

**The prompt for annotating region-level captions**

The full image is ⟨FULL_IMAGE⟩
The cropped object is at [x1, y1, x2, y2], ⟨CROP_IMAGE⟩

Analyze the given image and generate multiple detailed descriptions for the given object ('instance') found in the image. Each description must be accurate and unique, focusing solely on the information available in the image and annotations. Do not include any information that is not explicitly present in the image or annotation data. The descriptions should ensure that the object can be uniquely identified from a large set of similar images.

**Description Generation**:
1. Generate concise, clear descriptions.
2. Focus mainly on the object itself using:
- The object's inherent properties.
- The special details that can be used for separating other instances of the same category.
- Ensure each description allows the object to be uniquely identifiable within the image.
- Ensure diversity without referencing prior descriptions.
- Avoid direct mention of coordinate values.
3. For instances that are heavily occluded, blurry, or too small to be recognized due to a tiny bounding box, directly return "Instance quality is poor."

**Description Style**:
1. Use short sentences.
2. Minimize commas, avoid long or complex sentences.
3. Each description must reflect the interesting, accurate, and clear representation of the object, emphasizing the object as the focal point.
4. Each description should be more natural and aligned with human language conventions.
5. Each description must use the described object as the subject of the sentence.

Output the descriptions in JSON format.

---

---

**The prompt for annotating image-level captions**

The full image is ⟨FULL_IMAGE⟩

You are a precise, factual image cataloger. Your task is to generate a literal description of the image for a visual database. You should generate a **short caption** and a **long caption** for each image.

For short captions, follow these rules strictly:
1. **Identify Core Elements:** Describe the primary entities, objects, and the surrounding environment.
2. **Be Concise:** The entire description must be a single, clear sentence or phrase under 30 words.
3. **Be Natural:** Each description should be more natural and aligned with human language conventions.

For long captions, follow these rules strictly:
1. **Include Key Details:** Mention essential visual attributes like color, count, spatial relationships (e.g., "on the left", "in the background"), and relationships between objects.
2. **Be Objective:** Describe only what you can see. Strictly avoid any subjective language, atmosphere (e.g., "peaceful", "sad"), or interpretation of intent, actions, or the purpose of objects.
3. **Be Concise:** The entire description must be under 100 words but more than 50 words.
4. **Be Natural:** Each description should be more natural and aligned with human language conventions. Sentences need to be smooth and coherent.

Describe the image and output the descriptions in JSON format.

---

## C. Details of Local Image Retrieval Benchmarks

Local image retrieval is a challenging task in which the textual or visual query corresponds to only a small region or specific object within an image, rather than the entire scene. In this work, we evaluate on three established benchmarks:

SORCE-1K (Liu et al., 2025a) comprises 1,023 carefully curated images with complex backgrounds and textual queries that describe less prominent small objects with minimum surrounding context. The target objects typically occupy less than 10% of the image area, posing significant challenges for global image embedding models that focus on holistic scene understanding. Each query is associated with exactly one positive image, and performance is evaluated using Recall@1.

REIRCOCO (Hao et al., 2025) consists of 4,994 images from the COCO dataset, each annotated with a referring expression describing a specific object. The original evaluation protocol requires both image retrieval and object localization. However, since global embedding models lack localization capabilities, we adapt the protocol to a standard text-to-image retrieval setting, where the goal is to retrieve the correct image containing the described object. We report performance using Recall@1.

ILIAS (Kordopatis-Zilos et al., 2025) supports both text-based and image-based local retrieval, with queries formulated as a natural language description or a large image exemplar. The original dataset includes 5 million distractors, making full-scale evaluation computationally infeasible. Instead, we construct a manageable gallery using all 4,715 positive images. Different from the benchmarks mentioned above, ILIAS contains only 1,232 queries, each potentially matching multiple positive images. We evaluate using mAP@50, which computes the mean average precision over the top 50 retrieved results. Although our evaluation is conducted at a limited scale, we observe that the performance of ObjEmbed is quite robust to the number of distractors. As shown in Table 13, increasing the number of distractors leads to only a slight performance decline, and the overall ranking of methods remains unchanged.

*Table 13.* Effect of the number of distractors on ILIAS (T2I/I2I).

| #distractor | 0 | 5000 | 10000 |
|---|---|---|---|
| Qwen3-VL-Embedding-8B (Li et al., 2026) | 51.2/59.2 | 45.4/54.7 | 43.2/52.7 |
| ObjEmbed-4B | 77.6/85.3 | 74.1/83.3 | 72.1/82.1 |

# D. Visualization

**Visualizations of referring expression comprehension results.** In addition to strong performance on standard referring benchmarks (e.g., RefCOCO, RefCOCO+, RefCOCOg), in Figure 3, we demonstrate that ObjEmbed exhibits not only strong OCR capabilities (a, b, d), but also commonsense reasoning (c) and image-image matching abilities (e, f), highlighting its generalizability and versatility.

**Visualizations of local image retrieval results.** In Figure 4, we present qualitative results for three queries from the SORCE-1K dataset, showing the top-3 retrieved images. Our ObjEmbed not only ranks the correct target images as the top result but also accurately localizes the queried objects within the images. In contrast, even the state-of-the-art global image embedding model, Qwen-VL-Embedding-8B (Li et al., 2026), fails to retrieve the correct images in these challenging cases, highlighting the limitations of holistic representations in capturing fine-grained, localized visual content.

**Visualizations of self-annotated data.** Figure 5 shows representative examples from our self-annotated dataset. Thanks to carefully designed prompts and the use of frontier MLLMs, the generated captions are both accurate and highly distinctive.

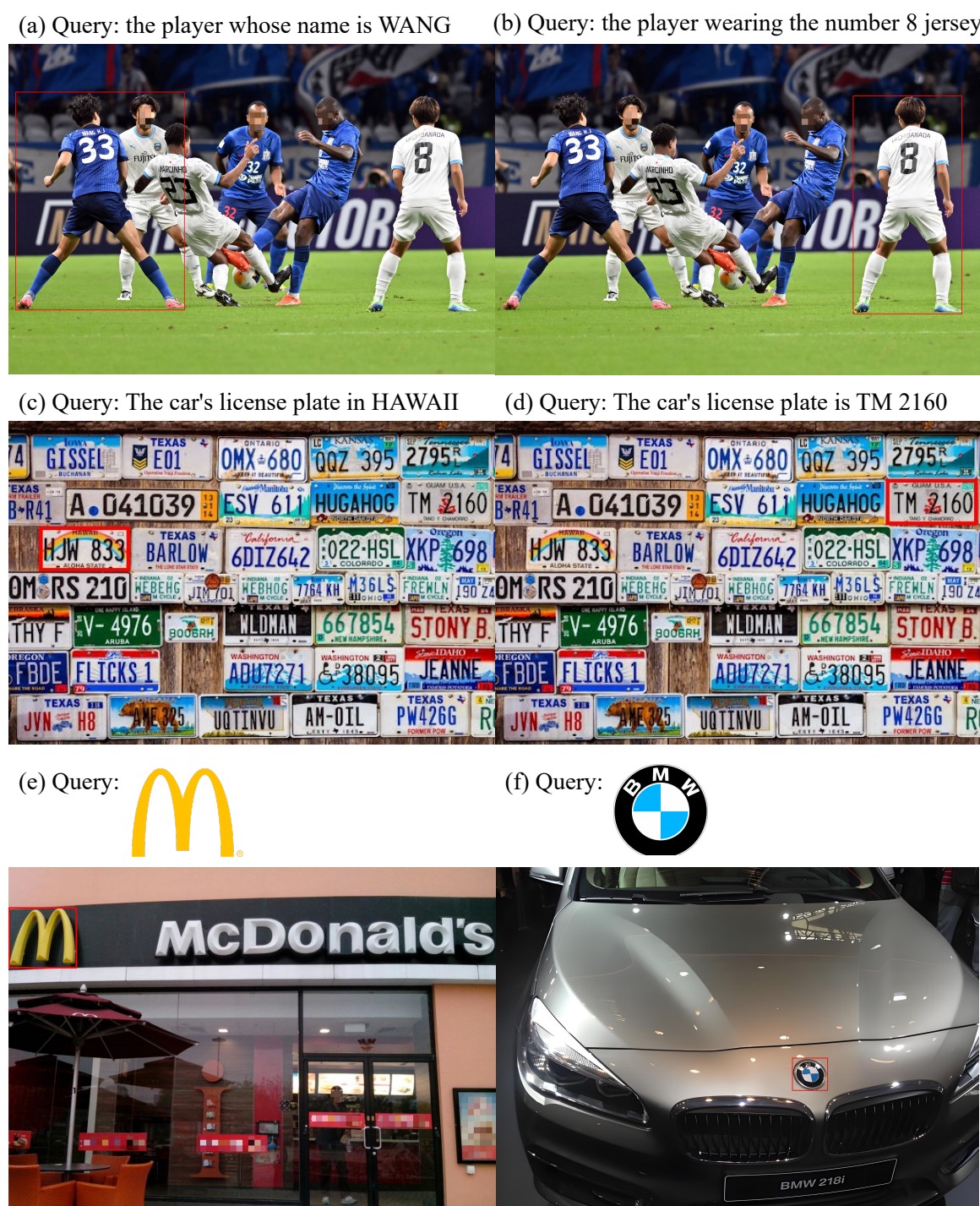

*Figure 3.* Visualizations of referring expression comprehension results with text queries and image queries.

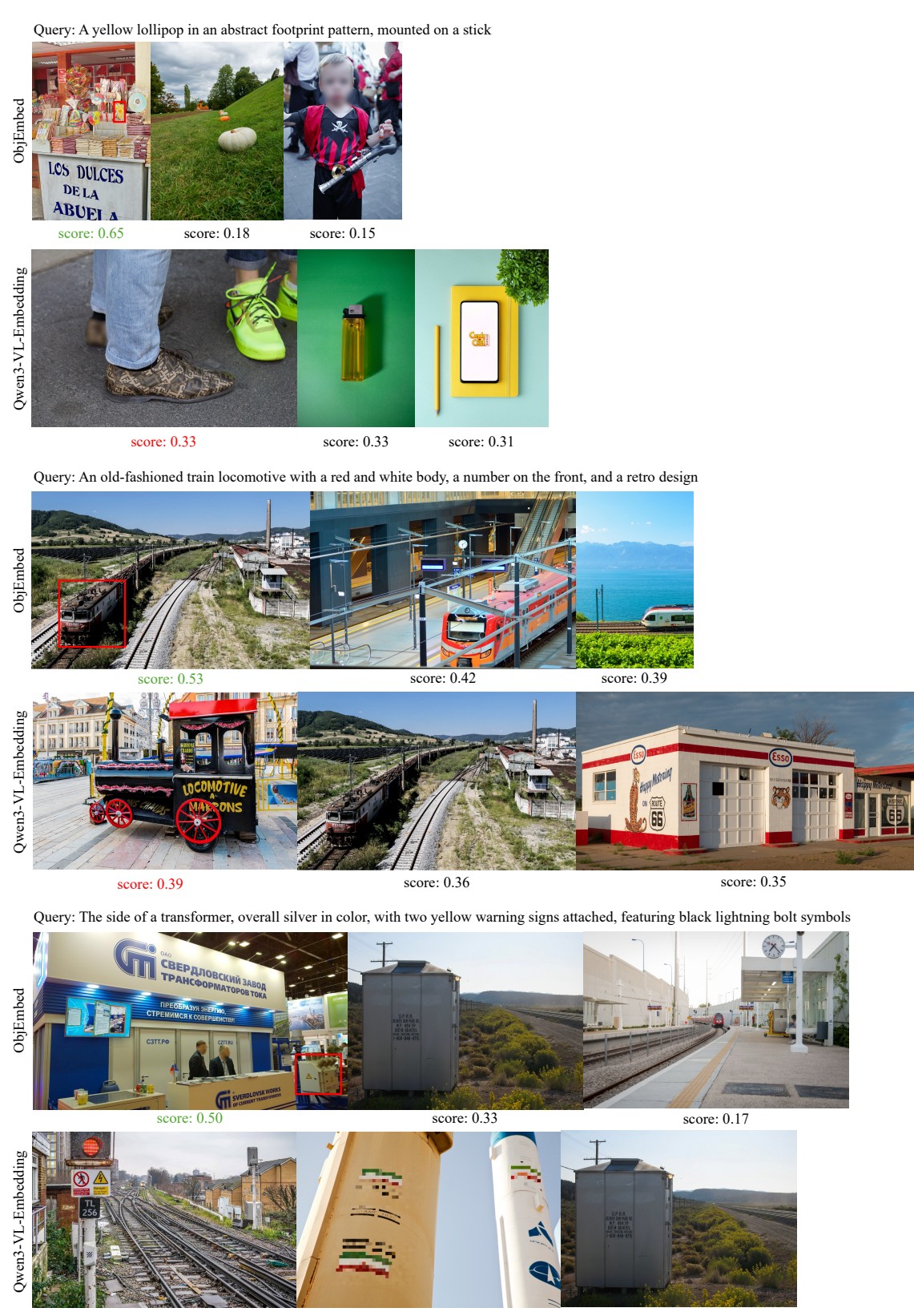

*Figure 4.* Visualizations of retrieval results on SORCE-1K. Our ObjEmbed successfully ranks the target image as the top result and accurately localizes the target objects (highlighted with red bounding boxes). In contrast, global image embedding models, like Qwen3-VL-Embedding 8B, tend to overlook small objects.

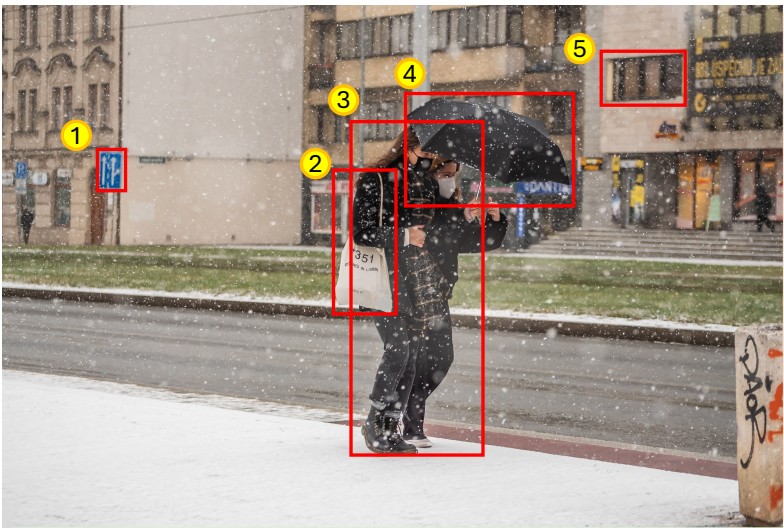

**short caption**
Two people in dark coats and face masks share a black umbrella while walking on a snowy sidewalk next to a city street.
**long caption**
Two individuals, both wearing dark coats and face masks, walk side-by-side on a snow-covered sidewalk. They share a single black umbrella. One carries a white tote bag with black text. Snow is visibly falling. In the background, a paved road, a grassy median, and multi-story buildings with storefronts are visible. A concrete bollard with graffiti is in the foreground on the right.

**region caption**

① A blue rectangular sign with two white arrows pointing straight ahead and one branching to the right.

② A white tote bag with black text reading +351 and DESIGNED IN LISBON is being carried by a person in a snowy scene.

③ A person wearing a black coat and dark pants holds a black umbrella while standing in falling snow.

④ A black umbrella is held aloft by two people standing close together in falling snow.

⑤ The window is dark and blurry with snowflakes falling in front of it.

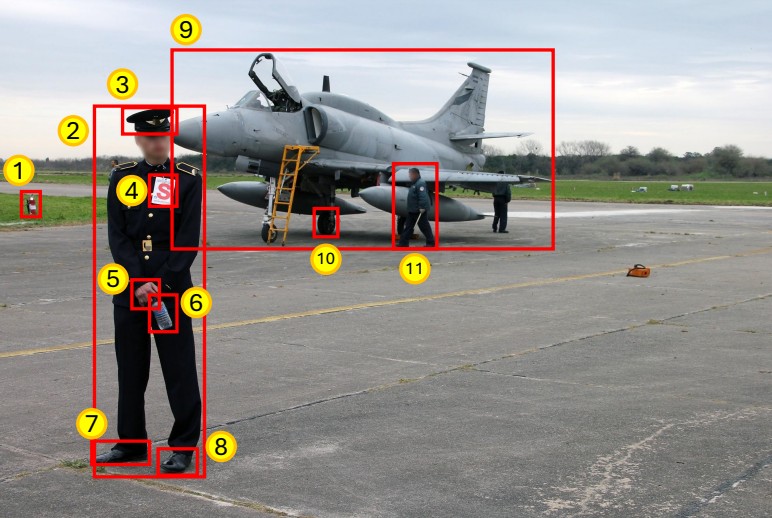

**short caption**
A uniformed officer stands on a tarmac near a gray military jet with its cockpit open, while other personnel work nearby.
**long caption**
A person in a dark military uniform with a peaked cap stands in the foreground on a paved airfield, holding a water bottle. Behind them, a gray single-engine jet fighter with its cockpit canopy open is parked. A yellow ladder leans against the aircraft, and at least two other individuals are visible near the plane. The background consists of a grassy field and a line of trees under an overcast sky.

**region caption**

① A red fire extinguisher with a black hose and handle is standing on a grassy area near an aircraft.

② A person in a dark military uniform with gold buttons and epaulets stands on a tarmac holding a water bottle and wearing a badge with a red S.

③ A dark military-style cap with a shiny metallic insignia centered on the front, including winged elements and a circular badge above.

④ A white security badge with a large red S and the word SEGURIDAD below it is pinned to a dark uniform.

⑤ A hand gripping a plastic water bottle with a blue cap.

⑥ A clear plastic water bottle with a blue label featuring red text is held in someone's hand.

⑦ A black dress shoe with a polished finish is stepping on a paved surface with a small patch of grass nearby.

⑧ A shiny black dress shoe with a polished toe cap is visible on a paved surface.

⑨ A gray military jet with its cockpit canopy open is parked on a tarmac with a yellow ladder positioned beside it.

⑩ The wheel is dark and circular with a visible rim and tire tread, partially obscured by shadow and low resolution.

⑪ A person in a dark jacket with a circular patch on the sleeve is walking while holding a long stick near a large aircraft.

*Figure 5.* Visualizations of self-annotated data. Each image is annotated with high-quality image-level and object-level captions. Images come from SA-1B (Kirillov et al., 2023).

