# OpenReview forum: "ObjEmbed: Towards Universal Multimodal Object Embeddings"
_ICML.cc/2026/Conference — ICML 2026 regular_

### Official Review · Reviewer_u2Le · 2026-02-18

**Soundness:** 3
**Presentation:** 4
**Significance:** 3
**Originality:** 2
**Overall Recommendation:** 5
**Confidence:** 4

**Summary:**

This paper, ObjEmbed, proposes a universal multimodal object embedding model that captures global and local objects as well as their text embeddings. The embeddings also focus on including information of bounding boxes, which is different than previous works.

**Compliance With Llm Reviewing Policy:**

Affirmed.

**Final Justification:**

My concerns have been adequately addressed, and I tend not to change my score.

**Key Questions For Authors:**

1. Why did the authors use the 2 caption templates "Find an ..."? Is it due to the downstream tasks? What if the model is trained on more templates or tested on different templates?
2. 2 Sigmoid focal losses are cited in sec 3.2; are they used differently in the two contrastive learning tasks? If so, the authors should probably refer to it differently.
3. It would be more interesting to show the comparison of ObjEmbed with more baseline models with sizes bigger than 4 B. Although the comparison might not be fair, it is still beneficial to see how close the proposed model is to these bigger models.
4. Since the model is trained on limited data (1.3M), how good is its generalization capability?

**Limitations:**

See the above discussion on novelty and baselines.

**Strengths And Weaknesses:**

## Strengths:
1. The writing is clear, and the motivation is too.
2. The authors explained all design choices and why so.

## Weaknesses:
1. Despite being tested thoroughly, the tasks the proposed method can deal with are not new, nor are the training losses. The innovation is the model architecture to include both local and global information, with bounding box information too. This, however, seems not too novel.
2. The authors should provide qualitative results on bounding boxes compared to the baseline models in the appendix.

#### minor
1. The author should also include the formula of "sigmoid focal loss" in the paper, despite already citing it.

---

> ### Author Rebuttal · Authors · 2026-03-30
>
> **W1: Clarify Novelty**
>
> **RE**: Please refer to the `W2 of Reviewer xk4a`.
>
> ---
>
> **W2: Qualitative Results**
>
> **RE**: We have included some visualizations in the Appendix. Since images cannot be uploaded directly to OpenReview, we will provide additional qualitative comparisons in the revised version of the paper.
>
> ---
>
> **W3 & Q2: The formula of "sigmoid focal loss"**
>
> **RE**: The sigmoid focal loss has the same mathematical form for both region-level and image-level contrastive learning. We cite different works, RetinaNet and SigLIP, because RetinaNet pioneered the use of sigmoid loss in object detection, while SigLIP applied it in image-text contrastive learning at the global (image-level) scale. The loss is defined as follows:
> $$
> \\begin{aligned}
>     &p_{i,j} = \\text{Sigmoid}(\\beta \\cdot s_{ij} + \\mu) \\\\
>     &FocalLoss(p_{i,j}, y_{i,j}) = \\begin{cases}
>         -\\alpha (1-p_{i,j})^\\gamma \\log(p_{i,j}), & \\text{if } y_{ij} = 1, \\\\
>         -(1-\\alpha) (p_{i,j})^\\gamma \\log(1-p_{i,j}), & \\text{if } y_{ij} = 0, \\\\
>     \\end{cases}
> \\end{aligned}
> $$
> where $s_{ij}$ is the cosine similarity between the object/image embedding $e^j$ and the text embedding $e^i$ defined in Equation 1. $\beta$ and $\mu$ are learnable and are not shared between region-level and image-level contrastive learning. $\alpha$ is set to 0.25 and $\gamma$ is set to 2 following common practice. For both tasks, we treat captions from all other GPUs within the same batch as negative samples during training. We will include these implementation details in the revised version of the paper.
>
> ---
>
> **Q1: Caption Template**
>
> **RE**: The templates are randomly selected. We use an LLM to rewrite the templates and find ObjEmbed is robust to them. Results are shown as follows.
>
> For object caption templates, we test on local retrieval benchmarks:
>
> - Find an object that matches the given caption. (68.5)
>
> - Identify the object in the image that corresponds to the provided caption. (68.6)
>
> - Locate the object that best matches the description given in the caption. (68.7)
>
> - Find the object that is most consistent with the given textual description. (68.5)
>
> - Determine which object in the image aligns with the provided caption. (68.7)
>
> - Select the object that accurately reflects the content of the given caption. (68.7)
>
> For image caption templates, we test on global retrieval benchmarks:
>
> - Find an image that matches the given caption. (81.7)
>
> - Retrieve an image that corresponds to the provided caption. (81.6)
>
> - Identify the image that best aligns with the given description. (81.6)
>
> - Locate the image that visually matches the provided textual caption. (81.6)
>
> - Select the image that accurately reflects the content of the given caption. (81.6)
>
> - Find the image that is most consistent with the described scene in the caption. (81.7)
>
> ---
>
> **Q3: Results of larger baseline models**
>
> **RE**: We have included several 7B–8B baseline models in Tables 3 and 4. Here, we further compare ObjEmbed-4B with Qwen3-VL-235B. It is important to note that Qwen3-VL-235B can not perform retrieval tasks and performs suboptimally in object localization. The official performance of Qwen3-VL-235B on the object detection benchmark ODinW13 is 48.6, while that of ObjEmbed-4B is 50.8. The performance of Qwen3-VL-235B on the REC benchmarks RefCOCO-avg is 91.9, while ours is 89.5. These results highlight the advantages of ObjEmbed in fine-grained perception tasks.
>
> ---
>
> **Q4: Model Generalization Capability**
>
> **RE**: First, although the dataset contains 1.3M images, it provides 8.1M bounding box annotations, offering rich and diverse supervision signals. Second, unlike CLIP, which is trained from scratch, MLLM-based embedding models inherit extensive world knowledge from base models, endowing them with superior generalization capabilities.
>
> We show the generalization ability of ObjEmbed from three aspects:
> **(1) Dataset Generalization**: We assess our model on 18 diverse benchmarks, 13 of which are evaluated in a zero-shot setting, meaning their training data is entirely excluded from our training set. The consistently strong performance across all benchmarks highlights ObjEmbed’s robust generalization to unseen datasets.
> **(2) Domain Generalization**: COCO-O includes images from six distinct non-natural domains (Cartoon, Hand-drawn, Painting, Sketch, Tattoo, and Weather), each significantly different from the natural images typically seen during training. ObjEmbed outperforms both traditional and MLLM-based methods by a clear margin, demonstrating its adaptability across diverse visual domains.
> **(3) Task Generalization**: As shown in Section 4.2, although ObjEmbed is trained exclusively on text-to-image (T2I) tasks, it surprisingly generalizes well to image-to-image (I2I) tasks, indicating its flexibility beyond the training paradigm.

---

> > ### Author Rebuttal · Reviewer_u2Le · 2026-03-31
> >
> > My concerns have been adequately addressed.

---

> > > ### Author Response · Authors · 2026-04-06
> > >
> > > We sincerely appreciate the reviewer's thoughtful comments and constructive feedback. We are grateful for the time and effort you have dedicated to reviewing our work. We are encouraged that all concerns have been adequately addressed, and we truly value your recognition of our work. Thank you once again for your support and valuable contribution to our research.

---

### Official Review · Reviewer_xk4a · 2026-03-05

**Soundness:** 3
**Presentation:** 3
**Significance:** 3
**Originality:** 3
**Overall Recommendation:** 4
**Confidence:** 4

**Summary:**

The paper proposes ObjEmbed, a novel Multimodal Large Language Model (MLLM) embedding architecture designed to generate both regional (object-level) and global image embeddings in a single forward pass. Instead of relying solely on global contrastive alignment, ObjEmbed utilizes an off-the-shelf region proposal network to extract regions of interest, which are then processed by the LLM as a sequence of tokens. A key contribution is the decoupling of object representation into two specialized tokens: an <object> token for semantic matching and an <iou> token to predict localization quality. The final matching score combines both semantic similarity and predicted localization quality. The authors evaluate the model across 18 benchmarks encompassing object detection, referring expression comprehension (REC), local image retrieval, and global image retrieval, demonstrating strong and versatile performance.

**Compliance With Llm Reviewing Policy:**

Affirmed.

**Key Questions For Authors:**

1. **Impact of the Distilled Training Dataset**: There is a critical lack of discussion regarding the impact of the newly curated dataset, which utilizes 1.3 million images and 8.1 million bounding boxes annotated via the massive Qwen3-VL-235B model. It is highly probable that this high-quality, distilled data from a 235B-parameter "teacher" model provides a substantial competitive advantage over the baselines, which do not have access to this generated information.
2. **Teacher Model Upper Bound**: To better contextualize the results and validate the effectiveness of the learning process from the annotated data, could the authors provide the performance of the teacher model (Qwen3-VL-235B)  on a subset of the key benchmarks? Demonstrating the teacher's performance would establish a clear upper bound and help illustrate how much of the original capability ObjEmbed successfully retains.

**Limitations:**

yes

**Strengths And Weaknesses:**

Strengths:
1. Extensive Empirical Validation: The paper conducts comprehensive experiments across 18 diverse benchmarks—spanning object detection, referring expression comprehension, and local/global image retrieval—achieving consistently strong and well-balanced results.
2. Clear and Well-Structured Presentation: The manuscript is exceptionally well-written. The structural organization is logical, making the multi-task learning objectives and the complex architectural designs easy to follow.

Weaknesses:
1. Lack of Explicit Mapping in Section 4.2: In the "Results on object detection benchmarks" section, the authors conclude that "Equipped with the dual-token design, ObjEmbed achieves a favorable balance between high semantic discriminability and precise location assessment". However, the manuscript lacks a clear, explicit mapping of the five evaluated datasets (COCO, COCO-O, ODinW13, FG-OVD, D3)  to these specific capabilities. Although there are hints earlier in the text regarding the limitations of traditional detectors versus MLLMs, explicitly stating which datasets primarily evaluate "semantic discriminability" and which strictly test "precise location assessment" would significantly improve readability and analytical depth.
2. Marginal Algorithmic Novelty : The proposed methodology leans heavily toward engineering and system integration rather than fundamental algorithmic or theoretical innovation.

---

> ### Author Rebuttal · Authors · 2026-03-30
>
> **W1: Object detection result analysis**
>
> **RE**: For COCO, COCO-O, and ODinW13, the text queries are typically category-level labels, such as "person". Each image in these datasets contains multiple instances of target objects, and performance is evaluated using the average precision (AP) metric with Intersection over Union (IoU) thresholds ranging from 0.5 to 0.95. As a result, accurate localization of individual objects is crucial. In contrast, the text queries in FG-OVD and D3 are significantly more detailed and descriptive, for example, "A dark green bench with a brown wooden back and seat and metal arms and legs". These fine-grained textual descriptions require models to precisely align multiple attributes, such as color, material, and structure, with the corresponding regions in the image, posing a greater challenge for accurate vision-language grounding. We will incorporate this analysis into the revised version of the paper.
>
> ---
>
> **W2: Marginal Algorithmic Novelty**
>
> **RE**: We highlight our novelty from two aspects:
>
> **Fundamental Paradigm Shift:**
>
> - ObjEmbed marks a shift from traditional global embedding models to local object-centric embedding models, offering distinct advantages in fine-grained visual understanding and local image retrieval. To the best of our knowledge, it is the first MLLM-based framework designed specifically for local object embedding.
>
> - ObjEmbed unifies multiple fine-grained vision tasks (object detection, REC, local image retrieval) within a single, unified object embedding space. To the best of our knowledge, ObjEmbed is the first model to seamlessly integrate these diverse tasks into a single framework, enabling versatile object-level understanding.
>
> **Technical Contributions:**
>
> - We introduce an efficient embedding mechanism that encodes all objects in a single forward pass, as opposed to processing each region individually, significantly improving computational efficiency.
>
> - We identify that localization quality is crucial for object representation, and thus explicitly model the IoU as a confidence measure for each embedded object, a key feature different from global embedding models.
>
> - We achieve a unified architecture that supports both local and global image embeddings without compromising performance on global image retrieval tasks, enabling comprehensive multi-level visual understanding within one model.
>
> ---
>
> **Q1: Impact of the Distilled Training Dataset**
>
> | Exp | Method | COCO | RefCOCO-avg | local retrieval avg | global retrieval avg |
> | :---: | :--- | :---: | :---: | :---: | :---: |
> | 1 | ObjEmbed-4B | 53.0 | 89.46 | 68.5 | 81.7 |
> | 2 | -self-collected data (500k) | 52.2 | 87.16 | 65.2 | 81.2 |
> | 3 | +FG-CLIP training data (700k) | 52.8 | 85.51 | 68.3 | 81.4 |
>
> **RE**: We would like to emphasize that the construction of a high-quality dataset is one of our contributions, and we plan to release this dataset publicly. As shown in the table above, without our self-collected data (ie, training with 800k data), the performance will degrade, especially in local image retrieval (Exp1 & Exp2). Additionally, integrating additional 700k training samples from FG-CLIP (ie, training with 2M data) leads to performance degradation on most benchmarks (Exp1 & Exp3), showing the high quality of our dataset.
>
> ---
>
> **Q2: Teacher Model Upper Bound**
>
> **RE**: We would like to highlight that Qwen3-VL-235B can not perform retrieval tasks and it is not good at object localization tasks. The official performance of Qwen3-VL-235B on the object detection benchmark ODinW13 is 48.6, while that of ObjEmbed-4B is 50.8. The performance of Qwen3-VL-235B on the REC benchmarks RefCOCO-avg is 91.9, while ours is 89.5. These results highlight the advantages of ObjEmbed in fine-grained perception tasks.

---

> > ### Author Rebuttal · Reviewer_xk4a · 2026-04-03
> >
> > My question has been resolved.

---

> > > ### Author Response · Authors · 2026-04-06
> > >
> > > We sincerely appreciate the reviewer's thoughtful comments and constructive feedback. We are grateful for the time and effort you have dedicated to reviewing our work. We are encouraged that all concerns have been adequately addressed, and we truly value your recognition of our work. Thank you once again for your support and valuable contribution to our research.

---

### Official Review · Reviewer_LPp7 · 2026-03-10

**Soundness:** 3
**Presentation:** 4
**Significance:** 4
**Originality:** 3
**Overall Recommendation:** 5
**Confidence:** 5

**Summary:**

This paper introduces ObjEmbed, a novel MLLM embedding framework designed to address the challenge of fine-grained alignment between image regions and their corresponding textual descriptions. Unlike previous models that focus mainly on global image-text alignment, ObjEmbed decomposes images into multiple regional embeddings, each representing an individual object as well as the whole image. The model generates two complementary embeddings per region: a semantic object embedding and an IoU embedding to estimate localization quality. ObjEmbed combines these to produce more accurate object-text matching. The framework is versatile, handling both region-level and image-level tasks efficiently in a single forward pass. Extensive experiments across 18 benchmarks show that ObjEmbed achieves superior performance in various visual understanding tasks.

**Compliance With Llm Reviewing Policy:**

Affirmed.

**Final Justification:**

Thanks to the author for the detailed response, which has given me a clearer understanding of the paper. I tend to maintain the current positive score.

**Key Questions For Authors:**

- Please provide a comparison of inference efficiency with existing methods across different tasks.
- Please include a more detailed description of the dataset construction process.
- How long was the model trained on 16 GPUs? Will you open-source the dataset and model?

**Limitations:**

yes

**Strengths And Weaknesses:**

## Strengths
- Unlike previous unified multimodal representation learning approaches, the proposed ObjEmbed can be effectively applied to a wide range of visual understanding tasks, including visual grounding, local image retrieval, and global image retrieval.
- The authors conduct extensive and repeated downstream experiments, which enhances the credibility of their results.
- The paper is well-written and easy to follow.

## Weaknesses
- This paper repeatedly emphasizes the efficiency of the proposed method; however, it lacks quantitative comparisons of inference efficiency with existing approaches across different downstream tasks.
- The paper lacks details on the sampling methods and dataset composition for the different datasets used in Table 1. Providing this information would be beneficial to the community.

---

> ### Author Rebuttal · Authors · 2026-03-30
>
> **W1 & Q1: Efficiency Analysis**
>
> **RE**: For retrieval tasks, ObjEmbed encodes all objects and the entire image in a single forward pass, similar to other global embedding models. ObjEmbed-4B runs at 6.9 fps, while Qwen3-VL-Embedding-8B runs at 9.5 fps, which are similar.
>
> For REC tasks, ObjEmbed requires two forward passes: one to compute object embeddings and another for text embeddings, with matching performed via dot product. As a result, ObjEmbed-4B achieves an inference speed of 4.0 fps. In contrast, our base model Qwen3-VL-4B must autoregressively decode coordinates token by token, leading to a significantly slower speed of 0.4 fps.
>
> For object detection, class text embeddings can be precomputed and shared across all images. Thus, ObjEmbed only needs a single forward pass to obtain object embeddings, enabling ObjEmbed-4B to run at 6.9 fps. Qwen3-VL-4B, however, needs even a few minutes for a sample due to the need to decode hundreds of tokens for multiple objects.
>
> ---
>
> **W2 & Q2: Please include a more detailed description of the dataset construction process**
>
> **RE**: We will open-source these data for clarity and reproduction. For all datasets used as our training data, we use their training split. For example, coco-train2017, lvis-v1-train, v3det-2023-v1-train, refcoco-train, FG-OVD-train, HumanRef-train, grefcoco-train, FineCops-Ref-train, REIRCOCO-train. For DAM data, we only use DAM-LVIS and DAM-OpenImages. For COCO-Family, we delete training images sharing the same names with test images.
>
> ---
>
> **Q3: How long was the model trained on 16 GPUs? Will you open-source the dataset and model?**
>
> **RE**: For the 4B model, it needs around 49 hours. For the 2B model, it needs around 35 hours. We will open-source the dataset and models.

---

> > ### Author Rebuttal · Reviewer_LPp7 · 2026-04-01
> >
> > My concerns have been adequately addressed.

---

> > > ### Author Response · Authors · 2026-04-06
> > >
> > > We sincerely appreciate the reviewer's thoughtful comments and constructive feedback. We are grateful for the time and effort you have dedicated to reviewing our work. We are encouraged that all concerns have been adequately addressed, and we truly value your recognition of our work. Thank you once again for your support and valuable contribution to our research.

---

### Official Review · Reviewer_NYJn · 2026-03-12

**Soundness:** 3
**Presentation:** 3
**Significance:** 2
**Originality:** 2
**Overall Recommendation:** 4
**Confidence:** 3

**Summary:**

ObjEmbed proposes an object-centric multimodal embedding model built by fine-tuning a large multimodal LLM (Qwen3-VL-Instruct) to output (i) a global image embedding (actually they output two global tokens, supervised by short vs. long captions); (ii) a set of object embeddings for region proposals. Each proposal is represented by two special-token embeddings: an object embedding intended for semantic matching, and an IoU embedding intended to predict the quality of the boxes; the model’s final object score combines semantic similarity with predicted localization quality. Training uses three objectives: (a) region-level supervision over proposals, using a sigmoid-focal objective, (b) image-level contrastive supervision on captions, and (c) IoU regression. The paper evaluates on set of tasks/benchmarks which comprises object detection, referring expression comprehension, local image retrieval, and global image-text retrieval, reporting strong and relatively balanced performance across these settings.

**Compliance With Llm Reviewing Policy:**

Affirmed.

**Final Justification:**

My primary concern with this paper was about reproducibility (no code was provided and details were a bit shady in my opinion), but after author reviewer discussion, the authors were allowed to link the code, and they motivated about my data quality concerns. I still think that while results are good, a lot of the work here is done by the region proposal network, on which the pipeline is dependant, and this dependance may need more experiments and analysis. However, I believe the work deserves consideration, thus I lean towards weak accept.

**Key Questions For Authors:**

Referring to some of the weaknesses :
1) how do you ensure no overlap between training images (including COCO-family sources and self-collected/crawled sets) and evaluation test splits for COCO retrieval, REIRCOCO, RefCOCO/g, etc.?
2) How sensitive are results to proposal recall/quality, number of proposals, and swapping WeDetect-Uni for other proposal sources?
3) What is the provenance and license status of the “self-crawled from licensed websites” images, and what portion (if any) will be released?
4) Please precisely define the image-level loss

**Limitations:**

Yes.

**Strengths And Weaknesses:**

Strengths
1) The method is well-designed and efficient: it encodes all object proposals and the global image context in a single forward pass, not needing autoregressive steps.

2) The proposed solution is simple and reusable. “Treat proposals as a sequence, read off special-token hidden states as embeddings” is conceptually clean and likely portable to other MLLMs.

3) Broad evaluation and ablations. The work evaluates on various benchmarks and includes ablations on token design, objectives, and supervision variants.

4) Decoupling semantics vs. localization via two tokens seems to work. In the ablation, the authors show that separating "what is it" and "how good is the box" improves performance and makes optimization better.

Major Weaknesses
1) The proposed approach is not an end-to-end object embedding model, as it has a fundamental dependence on an external proposal generator (WeDetect-Uni). This makes the system not a universal object embedding model in the usual sense. The model itself does not learn to identify or localize objects from scratch, and the authors explicitly note that incorporating bounding box regression actually degraded overall performance. Seems like the strong localization results on object detection benchmarks are largely the merit of the proposal generator recall capabilities. Given this, ObjEmbed functions more as a "second stage" contrastive aligner and semantic re-ranker for a predetermined set of candidates rather than a true universal object embedding model. For the same reason, as authors report in discussion, performance is necessarily bounded by proposal network recall and quality, and missing objects cannot be recovered nor the model can realize it is missing something. This dependency should be emphasized as a primary limitation and evaluated more rigorously (sensitivity to proposal quality, number of proposals, different proposal sources).
2) A large chunk of the training data (500k images) relies entirely on auto-generated annotations from another model, Qwen3-VL-235B. While the authors share the prompts they used, there is no mention of human verification, a secondary cross-check, or any real quality control pipeline. We all know that large multimodal models are prone to hallucinating or missing fine-grained details. Blindly trusting half a million auto-generated annotations without at least evaluating a small subset by hand introduces a huge risk of dataset noise.
3) Reproducibility is currently insufficient. There is no code/supplementary material beyond the appendix. In my opinion, given the complexity of the approach (proposal pipeline + fine-tuning + auto-annotation + evaluation adaptations), this is a serious weakness, especially since results may depend on implementation details (proposal scoring, NMS, prompt formats, split filtering). Region-level and IoU losses are well defined, but image-level contrastive details are comparatively vague (exact formulation, negatives, temperature/logit scaling, caption sampling across GPUs, etc.).

Minor Weaknesses
1) Potential training/evaluation contamination and unclear split hygiene. The training mixture includes datasets that overlap with evaluation benchmarks (e.g., COCO-family / REIRCOCO). Since there is no indication on which splits are used, it may be unclear if there is overlap in training/test data. I suggest to clearly indicate which split is used where.
2) For the local retrieval results, the paper adapts REIRCOCO to image retrieval and drastically reduces ILIAS distractors by constructing a smaller gallery. While these choices are reasonable for computational reasons, they change the task difficulty and complicate “SOTA” claims. The paper should clearly state this and provide some analysis to report how rankings change under closer-to-original protocols (even at reduced scale).

---

> ### Author Rebuttal · Authors · 2026-03-30
>
> | Exp | Method | COCO (AR/AP) | RefCOCO-avg (AR$_{50}$/Top1) | local retrieval avg | global retrieval avg |
> | :---: | :--- | :---: | :---: | :---: | :---: |
> | 1 | WeDetect-Uni (50 proposals) | 62.9/52.6 | 98.4/89.01 | 67.4 | 81.7 |
> | 2 | WeDetect-Uni (100 proposals) | 66.7/53.0 | 99.2/89.46 | 68.5 | 81.7 |
> | 3 | WeDetect-Uni (150 proposals) | 68.0/51.9 | 99.4/89.51 | 68.3 | 81.7 |
> | 4 | UPN (100 proposals) | 69.7/53.0 | 98.8/89.45 | 66.7 | 81.7 |
> | 5 | -self-collected data (500k) | 66.7/52.2 | 99.2/87.16 | 65.2 | 81.2 |
> | 6 | +FG-CLIP training data (700k) | 66.7/52.8 | 99.2/85.51 | 68.3 | 81.4 |
>
> **W1 & Q2: The model depends on an external proposal generator. How sensitive are results to proposal quality?**
>
> **RE**: Our primary goal is to learn universal object embeddings that can support multiple fine-grained vision tasks, such as object detection, REC, and local image retrieval, within a single, unified embedding space. Although our approach relies on a proposal network, constructing a versatile object-level embedding space remains challenging. To the best of our knowledge, ObjEmbed is the first framework to integrate these diverse tasks into a unified architecture. Moreover, MLLMs inherently struggle with precise bounding box regression in multi-object scenarios, as their next-token prediction mechanism operates in a classification-like manner and is not naturally suited to continuous coordinate regression. Therefore, it is a common paradigm that combines a proposal network with MLLMs to address object-centric tasks [e.g., ROD-MLLM (CVPR'25), ChatRex, RexSeek (ICCV'25), VLM-FO1, WeDetect-Ref (CVPR'26)].
>
> We find that ObjEmbed exhibits strong robustness to the choice of proposal network. As shown in the table (Exp1-4) above, ObjEmbed achieves similar performance across different numbers of proposals (50, 100, 150) and various proposal architectures (e.g., UPN, as used in ChatRex and RexSeek). The model effectively prioritizes high-quality proposals with higher matching scores, while suppressing low-quality ones. As a result, so long as the object recall is comparable, the final performance remains stable. Furthermore, the quality of proposals does not affect global retrieval results. We will include these analyses in the revised paper.
>
> ---
>
> **W2: Dataset quality**
>
> **RE**: To ensure data quality, we make efforts in 4 aspects: (1) using a strong model Qwen3-VL-235B; (2) annotating each object rather than the full image each time; (3) providing both the full image (context) and a cropped sub-image (detail) to the model during annotation; and (4) prompting the model to flag low-quality instances by outputting "Instance quality is poor."
>
> We manually check 100 images with 1,167 boxes and find only 13 box-error annotations and 57 label-error annotations. Without our self-collected data (ie, training with 800k data), the performance will degrade, especially in local image retrieval, as shown in the table above (Exp2 & Exp5). Additionally, integrating additional 700k training samples from FG-CLIP  (ie, training with 2M data) leads to performance degradation on most benchmarks (Exp2 & Exp6), showing the high quality of our dataset.
>
> ---
>
> **W3 & Q4: Provide more implementation details**
>
> **RE**: We will release our code and datasets for reproduction. We provide more details here and will include them in our revised paper. We select the top-100 object proposals without using proposal scoring. Prompts are listed in Appendix A. For image-level contrastive learning, we treat captions from all other GPUs within the same batch as negative samples, and the temperature parameter is learned end-to-end. Please refer to the `W3 of Reviewer u2Le` for the precise image-level loss.
>
> ---
>
> **W4 & Q1: Analysis on dataset overlapping**
>
> **RE**: For all datasets used as our training data, we use their training split. For all evaluation datasets, we use their validation or test split. For COCO-Family, we delete training images sharing the same filenames with test images. For self-collected data, we used DINOv2 to compute image similarities between them and all test images. Only 3 images exceed a similarity score of 0.99.
>
> ---
>
> **W5: Analysis on REIRCOCO and ILIAS**
>
> **RE**: For REIRCOCO, our ObjEmbed gets 32.96 BoxRecall@1(IoU>0.5), higher than the best performance 29.53 reported in the original paper.
>
> For ILIAS (T2I/I2I), we compare with the SOTA model Qwen3-VL-Embedding-8B with more distractors. In our setting (gallery size: 4715), ObjEmbed-4B gets 77.6/85.3 scores while Qwen gets 51.2/59.2. With 5000 more distractors, ObjEmbed-4B gets 74.1/83.3 scores while Qwen gets 45.4/54.7. With 10000 more distractors, ObjEmbed-4B gets 72.1/82.1 scores while Qwen gets 43.2/52.7. The performance of ObjEmbed degrades slightly and the rankings will not change.
>
> ---
>
> **Q3: Release dataset**
>
> **RE**: For all open-sourced images, we will release our annotations. For self-collected data, we will release them after passing the internal review.

---

> > ### Author Rebuttal · Reviewer_NYJn · 2026-04-02
> >
> > The authors have responded adequately to W1, Q2, Q1, and W4. However, my main concerns regarding data quality and reproducibility remain unresolved. The paper still does not provide enough detail to enable full reproduction of the experiments, and no supplementary code or implementation materials are currently available. In addition, the self-collected dataset remains insufficiently specified, and the quality of the 500k auto-labeled samples is still not convincingly validated.
> >
> > While such data may be useful in practice, it may also inherit and amplify the biases and limitations of the annotation model. The rebuttal mentions a manual inspection of 100 images, but it is unclear whether this analysis was already part of the original submission or was performed only in response to the reviews. In either case, I am not convinced that checking 100 images is sufficient to support claims about the quality of a dataset of 500k images. The rebuttal also does not explain how these 100 images were sampled, making it difficult to assess whether the estimate is meaningful or representative. Without a clearer validation protocol and stronger evidence of annotation quality, these concerns remain substantial.

---

> > > ### Author Response · Authors · 2026-04-06
> > >
> > > Thanks for your further comments and we are encouraged that we have addressed most of your technical concerns.
> > >
> > > **Regarding Reproducibility**
> > >
> > > We appreciate the reviewer's emphasis on reproducibility and take your concern seriously.
> > >
> > > Implementation details. We have provided comprehensive details in Section 4.1 and the Appendix, and have further supplemented them during the rebuttal. These details cover all major components of our method and the points raised in W3. We are happy to provide any further implementation details that the reviewer considers necessary for reproduction.
> > >
> > > Code availability. After consulting with the Program Chairs, we have been permitted to provide a code link. Here is our code (https://anonymous.4open.science/r/ObjEmbed-anonymous-8D22/README.md). It includes complete code for training, evaluation, and inference. We hope this facilitates full reproducibility of our results.
> > >
> > > **Regarding Data Quality**
> > >
> > > Regarding the data quality, we respectfully note that our dataset is not a standalone contribution but serves as training data to support our method. The standard for evaluating its quality should be whether it effectively enables the proposed approach, rather than the rigorous standards applied to dataset papers. Furthermore, Reviewer xk4a also agrees with us our dataset is high-quality in Q1. And we promise to open-source our dataset.
> > >
> > > We provide additional details to clarify our construction process and its impact on data quality. We highlight two aspects: (1) Use of a strongest annotator: Captioning is a well-performing capability of modern MLLMs, and the most advanced models, Qwen3-VL-235B, exhibit significantly lower hallucination rates. Generating annotations for the entire dataset required around 55k GPU hours.  (2) Quality control via self-monitoring: We instruct the model to explicitly flag instances of poor quality, such as those that are heavily occluded, blurry, or too small to recognize, by outputting "Instance quality is poor." This mechanism ensures that captions are only generated when the model is confident, thereby significantly enhancing reliability. During the process, around 4% of boxes were flagged and filtered out.
> > >
> > > We validate our data quality from **three aspects**:
> > >
> > > From the aspect of experimental validation, models trained with our dataset consistently improve performance across multiple tasks, while using FG-CLIP’s dataset leads to performance degradation under the same conditions, as shown in Exp5 and Exp6. You also acknowledge that such data may be useful in practice.
> > >
> > > From the aspect of human verification, a manual check reveals a low error rate. We provide the full protocol for the manual quality assessment. We **randomly** sampled 100 images from the dataset and employed two human annotators to examine all 1,167 bounding boxes individually. Each annotation was categorized into one of three classes: (1) Box Localization Error; (2) Text Label Error; (3) Correct. In cases of ambiguity, the annotators consulted as a group to reach a consistent judgment. The low number of annotation errors, only 13 box errors and 57 label errors, provides strong evidence of the high quality and reliability of our dataset. Given that the validation of 1,167 boxes required approximately half a day for two annotators, a full manual evaluation of the entire dataset is practically infeasible.
> > >
> > > | Dataset | #Images | #Objects | avg score | min score | max score |
> > > | :--- | :---: | :---: | :---: | :---: | :---: |
> > > | Ours | 300,000 | 3,161,694 | 0.268 | -0.040 | 0.540 |
> > > | FG-CLIP (ICML'25) | 300,000 | 1,015,615 | 0.222 | -0.195 | 0.497 |
> > >
> > > We further introduce an automated validation protocol for large-scale data assessment. Using CLIP-ViT-Large, we compute similarity scores between captions and cropped image regions. As shown in the table above, we compare our dataset with that of FG-CLIP, using a **randomly** selected subset of 300k images for evaluation. Our dataset contains three times more region-level annotations and achieves significantly higher CLIP similarity scores. Notably, the CLIP scores are highly consistent with human evaluation. In our manual verification, accurate boxes have an average CLIP score of 0.269, while incorrect annotations have a score of 0.249.
> > >
> > > We would like to highlight that our primary goal is to learn universal object embeddings that can support multiple fine-grained vision tasks. Our dataset is designed to enhance the discriminability of embeddings, and we demonstrate consistent performance improvements across multiple benchmarks. We acknowledge that the presence of a small amount of noise is inevitable in large-scale data and does not constitute a fundamental weakness of our approach. Instead, it reflects the practical challenges of scaling annotation efforts.
> > >
> > > We hope our responses provide clear and helpful clarification, and we appreciate the reviewer’s consideration during the evaluation process.

---

### Decision · Program_Chairs · 2026-04-30

**Decision:**

Accept (regular)

**Comment:**

All reviewers converged on a recommendation to accept. After carefully evaluating the reviews, rebuttal, discussion, and manuscript, the AC concurs with the assessment.